# Scalable Stochastic Gradient Riemannian Langevin Dynamics in Non-Diagonal Metrics

**Hanlin Yu**                                                                              *hanlin.yu@helsinki.fi*
*Department of Computer Science, University of Helsinki, Finland*

**Marcelo Hartmann**                                                          *marcelo.hartmann@helsinki.fi*
*Department of Computer Science, University of Helsinki, Finland*

**Bernardo Williams**                                          *bernardo.williamsmoreno@helsinki.fi*
*Department of Computer Science, University of Helsinki, Finland*

**Arto Klami**                                                                              *arto.klami@helsinki.fi*
*Department of Computer Science, University of Helsinki, Finland*

**Reviewed on OpenReview:** *https: // openreview. net/ forum? id= dXAuvo6CGI*

## Abstract

Stochastic-gradient sampling methods are often used to perform Bayesian inference on neural networks. It has been observed that the methods in which notions of differential geometry are included tend to have better performances, with the Riemannian metric improving posterior exploration by accounting for the local curvature. However, the existing methods often resort to simple diagonal metrics to remain computationally efficient. This loses some of the gains. We propose two non-diagonal metrics that can be used in stochastic-gradient samplers to improve convergence and exploration but have only a minor computational overhead over diagonal metrics. We show that for fully connected neural networks (NNs) with sparsity-inducing priors and convolutional NNs with correlated priors, using these metrics can provide improvements. For some other choices the posterior is sufficiently easy also for the simpler metrics.

## 1 Introduction

Bayesian methods are increasingly being used for large-scale models, especially deep neural networks, with significant literature on both sampling-based algorithms (Welling & Teh, 2011; Chen et al., 2014; Zhang et al., 2020; Vono et al., 2022) and distributional approximation algorithms (Graves, 2011; Blundell et al., 2015; Huix et al., 2022; Nazaret & Blei, 2022). Even though sampling-based methods are often considered inefficient in the general Bayesian inference literature, they are highly practical for neural networks due to the stochastic-gradient sampling methods following closely the process of stochastic optimization (Wenzel et al., 2020). In effect, we get efficient samplers by relatively minor modification of optimization algorithms. There are two prominent families of stochastic-gradient sampling methods, based on Langevin dynamics (Welling & Teh, 2011) and Hamiltonian dynamics (Chen et al., 2014). We build on the former that has the advantage of ease of implementation and fewer hyper-parameters, but our contributions focusing on accounting for local curvature could likely be used also in Hamiltonian dynamics.

The samplers based on Langevin dynamics operate by numerical integration of a particular stochastic differential equation (SDE) (Ma et al., 2015). Our starting point is the Stochastic Gradient Riemannian Langevin Dynamics (SGRLD) method where the underlying SDE is defined on a *manifold* characterised by a metric tensor $G(\boldsymbol{\theta}) \in \mathbb{R}^{D \times D}$ (for a problem of $D$ parameters) and the integration is carried along the manifold. This formulation allows improved sampling by careful choice of the metric (Ma et al., 2015). Perhaps the optimal

choice is to use metrics that accurately capture the local curvature of the target distribution and e.g. Girolami & Calderhead (2011) have demonstrated that using the Fisher information matrix as the metric tensor provides advantages in exploration for sufficiently small problems where it is technically feasible. However, this comes with a substantial computational challenge: Already constructing the Fisher information matrix is computationally costly and the samplers need to repeatedly compute its inverse and inverse square root.

For large-scale problems the practical methods sacrifice flexibility for computational efficiency by using simplified metrics. The early works used constant $G$ that was further restricted to be diagonal, resulting in the basic SGLD method (Welling & Teh, 2011). More recent methods use position-dependent $G(\boldsymbol{\theta})$ but assume it to be diagonal to retain computational efficiency. In a suitably chosen diagonal metric the computational overhead is negligible as element-wise operations can be used throughout the algorithm. Furthermore, the metric is typically not formed using differential geometric arguments, and is often adaptively tuned during optimization. For example, Li et al. (2016) used $G(\boldsymbol{\theta})$ motivated by the RMSprop preconditioner used commonly for optimization, and Wenzel et al. (2020) constructed a diagonal $G(\boldsymbol{\theta})$ with separate scaling for each layer of a neural network. These methods remain the standard choices for inference for general cases. Some attempts of using non-diagonal metrics have been done, but all of the current solutions are structure-dependent and only applicable for specific scenarios. Marceau-Caron & Ollivier (2017) and Nado et al. (2018) provide computationally more efficient approximations for the Fisher information metric in form of quasi-diagonal approximations and the K-FAC approximation, respectively, but they still incur a higher computational cost and require explicit construction of the approximation for a specific network structure. Recently, Lange et al. (2023) constructed a non-diagonal preconditioner that accounts for the curvature induced by the batch normalization operation and showed that it improves efficiency for networks using that operation, but they do not provide a general recipe for forming the metric for all other structures.

In this work we provide two new SGRLD variants using non-diagonal metrics $G(\boldsymbol{\theta})$ that – when implemented in a suitable manner – are computationally efficient and applicable for general deep neural networks with arbitrary network structures. The resulting algorithms yield well-defined operations for different network layer types without specific treatments, in contrast to the previous attempts. The first method builds on the *Monge* metric Hartmann et al. (2022) introduced in the context of Lagrangian Monte Carlo (Lan et al., 2015). This metric is a combination of a diagonal matrix and a rank-one matrix that depends only on the gradients, taking the form $I_D + \alpha^2 \nabla\ell\nabla\ell^\top$ with $\ell$ being the log posterior, having a differential geometric justification. Despite of full-matrix form, the metric is fast to compute and has efficient inverse. We incorporate the metric into SGRLD and show that we can implement many operations in element-wise manner just as in the case of diagonal metrics. The sampler is generally applicable as the metric does not rely on e.g. the structure of a neural network, but it is found to be quite sensitive to a tuning parameter $\alpha^2$ controlling the metric.

The second metric builds on the Shampoo method Gupta et al. (2018) proposed for optimization. They construct a full matrix preconditioner $G(\boldsymbol{\theta})$ by using Kronecker products of layer-wise matrices that can be efficiently updated during optimization. We incorporate their preconditioner as a metric tensor in SGRLD. The updates are considerably more efficient than direct inversion of the full matrix but the sampler still has some computational overhead compared to diagonal metrics, in our experiments around factor of $1.3 - 2.4$. We argue that such small overhead is justified by consistent good performance in our experiments.

We evaluate the metrics in problems of varying complexity. We observe that for fully connected neural networks with Gaussian priors the metric may not be that important. For heavy-tailed priors often recommended for these networks (Fortuin et al., 2022), such as the horseshoe prior (Carvalho et al., 2009) used by Ghosh et al. (2019) and Popkes et al. (2019), the posterior is more challenging and the choice of the metric is more important. For convolutional neural networks, we also observe differences between the metrics.

## 2   Background

We denote by $\boldsymbol{\theta} \in \mathbb{R}^D$ the parameters of a model, here a neural network, and want to infer the posterior

$$p(\boldsymbol{\theta} \,|\, X) = \frac{\exp(-U(\boldsymbol{\theta})/\tau)}{Z} \tag{1}$$

expressed in terms of log-potential $U(\boldsymbol{\theta}) = -\log p(X, \boldsymbol{\theta})$ to facilitate usage of gradient-based algorithms. Here $Z$ is the intractable normalization constant and $\tau$ is a temperature parameter that is sometimes used to improve predictive performance (Wenzel et al., 2020). We write the equations in this general form but use $\tau = 1$ corresponding to standard Bayesian inference in all experiments. We use the notation $\hat{x}$ for a stochastic estimate for a quantity $x$, estimated from a subsample of observations. In particular, $\nabla_{\boldsymbol{\theta}}\hat{U}(\boldsymbol{\theta})$ is the estimate of the full gradient $\nabla_{\boldsymbol{\theta}}U(\boldsymbol{\theta})$ and we refer to the estimate as the stochastic gradient.

Stochastic Gradient Riemannian Langevin Dynamics (SGRLD) (Girolami & Calderhead, 2011; Patterson & Teh, 2013; Ma et al., 2015) builds on the Stochastic Differential Equation (SDE) (Särkkä & Solin, 2019)

$$d\boldsymbol{\theta} = -G(\boldsymbol{\theta})^{-1}\nabla_{\boldsymbol{\theta}}U(\boldsymbol{\theta})dt + \sqrt{2\tau}G(\boldsymbol{\theta})^{-\frac{1}{2}}dW + \tau\Gamma(\boldsymbol{\theta})dt, \tag{2}$$

where

$$\Gamma_j(\boldsymbol{\theta}) = \sum_{k=1}^{D} \frac{\partial}{\partial\boldsymbol{\theta}_k}(G(\boldsymbol{\theta})^{-1})_{jk}.$$

Here $W$ is Brownian motion, $G(\boldsymbol{\theta}) \in \mathbb{R}^{D \times D}$ is a metric tensor characterising the manifold we are operating on, and the remaining parameters correspond to those of Equation 1. The SDE has a differential geometric justification, as it can be seen as performing sampling on the manifold (Girolami & Calderhead, 2011). With $G(\boldsymbol{\theta}) = I_D$, we recover the original SGLD proposed by Welling & Teh (2011). Under some mild conditions, it can be proven that we obtain approximate samples from the posterior distribution by following the trajectories induced by the SDE (Ma et al., 2015). In principle any integrator could be used, but in practice the Euler-Maruyama integrator is used throughout the SGLD literature. The SDE is discretized with step-size $h_t > 0$ and the samples are obtained by iterating the update rule using stochastic gradients

$$\boldsymbol{\theta}_{t+1} = \boldsymbol{\theta}_t - G(\boldsymbol{\theta}_t)^{-1}\nabla_{\boldsymbol{\theta}_t}\hat{U}(\boldsymbol{\theta}_t)h_t + \sqrt{2\tau h_t}G(\boldsymbol{\theta}_t)^{-\frac{1}{2}}\mathcal{R}_t + \tau\Gamma(\boldsymbol{\theta}_t)h_t, \tag{3}$$

where $\mathcal{R}_t \sim \mathcal{N}(0, I_D)$. The total integration time becomes $S_T = \sum_{t=1}^{T} h_t$ where $T$ is the number of iterations.

This integrator has two core computational challenges. The second and third terms require computing the metric's inverse and square root of the inverse, which in general has $O(D^3)$ complexity. The other fundamental challenge is that in the last term $\Gamma(\boldsymbol{\theta})$ involves derivatives of the inverse and is challenging even if the inverse itself could be obtained efficiently. However, that term can luckily be omitted in practical algorithms. Girolami & Calderhead (2011) showed that the term naturally disappears with the assumption that the metric tensor does not depend on the model parameters but is constant $G(\boldsymbol{\theta}) = M$. Moreover, Li et al. (2016) provides the following theorem that indicates we get bounded estimation error for quantities of interest even if completely ignoring the term, and consequently we can derive practical samplers by not considering $\Gamma(\boldsymbol{\theta})$ explicitly as long as we verify we have a similar bound for any given metrics we consider.

**Theorem 2.1.** *Denote operator norm as $\|\cdot\|$. With assumption on smoothness and boundedness as provided in the Appendix A.1, after ignoring the $\Gamma(\boldsymbol{\theta})$ terms during discretized updates, for any quantity $\phi$ estimated as the empirical expectation over the posterior samples, the mean-square error of the estimate for a general SGRLD sampler is bounded as*

$$E\left(\hat{\phi} - \bar{\phi}\right)^2 \leq C\left(\sum_t \frac{h_t^2}{S_T^2}E\|\Delta V_t\|^2 + \frac{1}{S_T} + \frac{(\sum_{t=1}^{T} h_t^2)^2}{S_T^2} + \left\|\sum_{t=1}^{T} \frac{h_t}{S_T}\tau\Gamma(\boldsymbol{\theta}_t)\right\|^2\right)$$

*for some constant $C > 0$ independent of $\{h_t\}$, where $\Delta V_t$ is an operator that is defined in the Appendix A.1 where we also replicate the proof in our notation.*

## 2.1 Previous SGRLD methods

Most previous SGRLD methods build on Euler-Maruyama integration as in Equation 3, only differing in the choice of the metric. Next we briefly explain the key methods commonly used in practice from the perspective of the metric, always presenting both the metric and the inverses required for performing the integration to highlight their computational properties. Importantly, all of these use diagonal metrics. To simplify notations, we will use $\hat{g}$ to denote $\nabla_{\boldsymbol{\theta}}\hat{U}(\boldsymbol{\theta})$ divided by the number of data points in the training set.

**SGLD**   Welling & Teh (2011) used constant metric

$$G(\boldsymbol{\theta}_t) = G(\boldsymbol{\theta}_t)^{-1} = G(\boldsymbol{\theta}_t)^{-\frac{1}{2}} = I_D. \tag{4}$$

This is highly efficient, but the metric is completely naive.

**pSGLD**   Li et al. (2016) constructed $G(\boldsymbol{\theta})$ inspired by the RMSprop preconditioner. We formulate it as

$$
\begin{aligned}
G(\boldsymbol{\theta}_t) &= \mathrm{diag}\left(v(\boldsymbol{\theta})\right), \\
G(\boldsymbol{\theta}_t)^{-1} &= \mathrm{diag}\left(1 \oslash v(\boldsymbol{\theta})\right), \\
G(\boldsymbol{\theta}_t)^{-\frac{1}{2}} &= \mathrm{diag}\left(1 \oslash \sqrt{v(\boldsymbol{\theta})}\right),
\end{aligned}
$$

where $V(\boldsymbol{\theta})$ is initialized as a zero vector and updated with exponential moving average (EMA) with suitable $\lambda$ as

$$
\begin{aligned}
V(\boldsymbol{\theta}_t) &= \lambda V(\boldsymbol{\theta}_{t-1}) + (1-\lambda)\hat{g}^2, \\
v(\boldsymbol{\theta}_t) &= \sqrt{V(\boldsymbol{\theta}_t)} + \epsilon.
\end{aligned}
$$

The notation $\mathrm{diag}(\cdot)$ denotes the operation that generates a diagonal or block-diagonal matrix from an arbitrary number of scalars or matrices, and $\epsilon$ is a small constant added mainly for numerical reason. Here the square $\cdot^2$, square root $\sqrt{\cdot}$ and division $\oslash$ are all performed element-wise and hence the algorithm has linear cost; we consistently use the notation where $\sqrt{\cdot}$ refers to element-wise operation and the matrix square root is denoted by $G^{1/2}$.

**Wenzel's SGLD**   Wenzel et al. (2020) constructed a layer-wise metric. The implementation by Fortuin et al. (2021) follows

$$
\begin{aligned}
G(\boldsymbol{\theta}_t) &= \mathrm{diag}\left(\{\tilde{\sigma}(\boldsymbol{\theta}_t)_l I_{D_l}\}_{l=1}^{L}\right), \\
G(\boldsymbol{\theta}_t)^{-1} &= \mathrm{diag}\left(\{(1/\tilde{\sigma}(\boldsymbol{\theta}_t)_l)\, I_{D_l}\}_{l=1}^{L}\right), \\
G(\boldsymbol{\theta}_t)^{-\frac{1}{2}} &= \mathrm{diag}\left(\left\{\left(1/\sqrt{\tilde{\sigma}(\boldsymbol{\theta}_t)_l}\right) I_{D_l}\right\}_{l=1}^{L}\right),
\end{aligned}
$$

where $L$ is the number of layers and $D_l$ the number of parameters for the $l$th layer, and

$$
\begin{aligned}
\sigma(\boldsymbol{\theta}_t)_l &= \sqrt{\epsilon + \mathrm{mean}(V([\boldsymbol{\theta}_t]_l))}, \\
\tilde{\sigma}(\boldsymbol{\theta}_t)_l &= \sigma_l / \min(\{\sigma(\boldsymbol{\theta}_t)_l\}_{l=1}^{L}),
\end{aligned}
$$

where the update rule of $V([\boldsymbol{\theta}_t]_l)$ is similar to above, but initialized to be a vector full of ones. $\epsilon$ has a similar role to above and the operations are element-wise. The preconditioner is updated periodically, in practice after every epoch.

## 3   SGRLD in Non-diagonal metrics

For more efficient exploration of the posterior, for instance when the parameters are strongly correlated or the posterior involves areas of strong curvature, we should be using full matrix $G(\boldsymbol{\theta})$, rather than a diagonal one that can only scale the dimensions. Even though the computation for arbitrary metrics is prohibitively expensive, we can construct specific metrics of low cost that still capture the correlations.

Next we introduce two such metrics that lead to efficient SGRLD samplers without assuming the metric to be diagonal. The first is based on a metric derived from the perspective of differential geometry that has efficient inverses due to being a combination of a diagonal matrix and a rank-one term, proposed by (Hartmann et al., 2022) for sampling algorithms with exact gradients. The latter uses efficient Kronecker products to construct a metric that is sufficiently efficient to compute for large neural networks, originally proposed in the optimization context (Gupta et al., 2018; Anil et al., 2021) and generalized here for sampling.

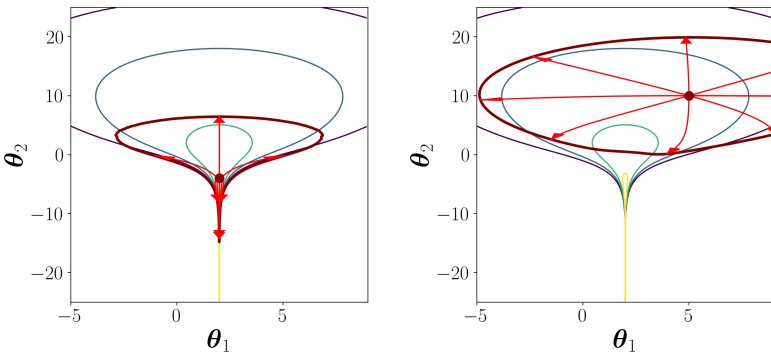

Figure 1: Illustration of how the Monge metric captures the local curvature of the target density, here the funnel distribution (see Section 4.2). The two plots illustrate the local metric around two separate parameter settings, in form of geodesic paths (red arrows) for different initial velocities sampled from a Euclidean ball and the surfaces of the final positions (solid red line). The metric helps in reaching the narrow funnel in y-direction (left) and is similar to Euclidean metric in the flat areas (right).

### 3.1 SGRLD in Monge metric

Hartmann et al. (2022) recently proposed a metric named *Monge metric*. They embed the target probability distribution in a higher-dimensional Euclidean space as $\mathcal{M} = \xi(\Theta)$, where $\boldsymbol{\theta} \xrightarrow{\xi} (\boldsymbol{\theta}, \alpha U(\boldsymbol{\theta}))$, yielding a first fundamental form on $\mathcal{M}$, which can be seen as a natural metric induced by the graph of the function. It thus has a direct differential geometric justification. Figure 1 illustrates the geodesics induced by the metric for the funnel distribution, demonstrating how it characterises the local curvature. See Section 4.2 for details of the example and empirical validation of how SGRLD in the Monge metric, as described next, also improves sampling from funnel.

The metric is given by $G(\boldsymbol{\theta}) = I + \alpha^2 \nabla_{\boldsymbol{\theta}} U(\boldsymbol{\theta}) \nabla_{\boldsymbol{\theta}} U(\boldsymbol{\theta})^\top$ and hence has the inverse (dropping dependence on $\boldsymbol{\theta}$ for lighter notation) $G(\boldsymbol{\theta})^{-1} = I - \frac{\alpha^2}{1+\alpha^2\|\nabla U\|^2}\nabla U \nabla U^\top$ and equally easy square root of that. Hartmann et al. (2022) used it in the context of Lagrangian Monte Carlo (Lan et al., 2015) and showed that the metric helps to achieve good exploration of complex target distributions, but their sampler does not scale for large problems due to requiring exact gradients and determinants for rejection checks.

Here we use that metric in the context of SGRLD, which requires some additional development. Even though the expressions above are efficient compared to direct matrix inversion, the cost of actually forming the square metric from the direct application of Sherman-Morrison inversion would still be a challenge for very large models. Next we show that by careful manipulation of the equations we avoid forming the matrix explicitly and can implement SGRLD with element-wise operations, eventually matching the computational cost of diagonal approximations. We also show that standard exponential moving average of stochastic gradients is applicable for forming the metric tensor.

To derive the efficient element-wise operations, we start by defining helper functions to simplify notations:

$$f_{-1}(x) = -\frac{\alpha^2}{1 + \alpha^2\|x\|^2},$$

$$f_{-\frac{1}{2}}(x) = \frac{1}{\|x\|^2}\left(\frac{1}{\sqrt{1 + \alpha^2\|x\|^2}} - 1\right).$$

We can then express the metric $G(\boldsymbol{\theta})$, its inverse and inverse square root as

$$
\begin{aligned}
G(\boldsymbol{\theta}_t) &= I_D + \alpha^2 \nabla l_t \nabla l_t^\top, \\
G(\boldsymbol{\theta}_t)^{-1} &= I_D + f_{-1}(\nabla l_t) \nabla l_t \nabla l_t^\top, \\
G(\boldsymbol{\theta}_t)^{-\frac{1}{2}} &= I_D + f_{-\frac{1}{2}}(\nabla l_t) \nabla l_t \nabla l_t^\top,
\end{aligned}
\tag{5}
$$

where $\nabla l_0$ is initialized to zero and updated based on the stochastic gradient using standard EMA as

$$\nabla l_t = \lambda \nabla l_{t-1} + (1 - \lambda)\hat{g}_t.$$

Observe that all computations required by the integrator are of the form $G(\boldsymbol{\theta})^n x$ for some vector $x$ and either $n = -1$ or $n = -1/2$. Using this notation, the $l$th entry of the required product can be computed using

$$[G(\boldsymbol{\theta}_t)^n x]_l = [x + f_n(\nabla l_t)\nabla l_t \langle \nabla l_t, x \rangle]_l$$
$$= [x]_l + f_n(\nabla l_t)[\nabla l_t]_l \langle \nabla l_t, x \rangle,$$

where $\langle \cdot, \cdot \rangle$ is the inner product that combines the last element of Equation 5 with $x$. Since $f(\cdot)$ only depends on the current moving average of gradients, we can implement the exact updates for a batch by iterating through all the parameters once to calculate the needed quantities, followed by individual updates for each of the parameters. In practice, all operations are still computed with standard tensor operations.

Both the memory and computation complexity is linear in $D$, which is the same as using diagonal precon­ditioners like RMSprop as in Li et al. (2016). However, it is important to note that the metric does not correspond to a diagonal approximation. As shown by Equation 5, the effect of the metric is *additive* instead of *multiplicative*. For all diagonal samplers the $G(\boldsymbol{\theta})^n x$ terms in Equation 3 take the forms where elements of the gradient are multiplied by a scaling factor, whereas here we retain the gradient itself and add a scaled version of moving average to it.

For the specific case of probabilistic models, i.e. $U(\boldsymbol{\theta}) = -\log \pi_Y(\boldsymbol{y}|\boldsymbol{\theta})$, the Fisher information matrix characterizes the local curvature. As shown by Hartmann et al. (2022), Monge metric relates to the Fisher information matrix in expectation (with respect to $Y$) such that its expected value is identity matrix plus $\alpha^2$ times the Fisher information matrix. However, this connection is somewhat theoretical as in practical use we evaluate it for the observed data $y$ rather than computing the expectation. It is also important to note that despite similar expressions the outer product used for constructing the Monge metric does not correspond to the empirical estimate of Fisher information discussed by Kunstner et al. (2019). The Monge metric uses the outer product of full-data gradients (or their stochastic estimates), whereas the empirical Fisher estimate is an expectation of the outer products of per-sample gradients, as detailed Appendix A.8. The expectation of the outer product has different computational properties than the outer product of expectations, and the observations Kunstner et al. (2019) make do not directly apply. However, the Monge metric retains the problematic scaling of empirical Fisher due to use of "squared" gradients, and the scaling parameter $\alpha^2$ is needed in part to account for this.

We will later observe that the performance depends strongly on the choice of $\alpha^2$ and hence the metric introduces a new hyper parameter that needs to be selected carefully. When $\alpha^2 = 0$ the metric reduces to identity and differs increasingly more from the Euclidean one for larger values. Note that $\alpha$ value here does not have the same interpretation as in Lagrangian Monte Carlo, as there is an implicit scaling depending on the number of data points in the training set. The validity of the sampler for all $\alpha^2 \geq 0$ is shown by the following theorem, with the proof provided in the Appendix A.2.

**Theorem 3.1.** *For SGRLD with the Monge metric, after ignoring the $\Gamma(\boldsymbol{\theta})$ terms during discretized updates, we can bound the approximation error as defined in Theorem 2.1 as*

$$E\left(\hat{\phi} - \bar{\phi}\right)^2 \leq C \left( \sum_t \frac{h_t^2}{S_T^2} E\|\Delta V_t\|^2 + \frac{1}{S_T} + \frac{(\sum_{t=1}^{\top} h_t^2)^2}{S_T^2} \right) + O(\alpha^8 \tau^2 (1 - \lambda)^2).$$

We note that the bound is provided solely for ensuring asymptotic validity of the sampler and it does not directly say anything about the empirical accuracy of the sampler for different $\alpha$. There is no reason to believe the bound to be particularly tight and the large exponent in $\alpha^8$ is not necessarily problematic as already a small $\alpha$ can dramatically change the metric.

### 3.2 SGRLD in Shampoo metric

An alternative to using a metric with efficient inverses is to use a metric for which the inverses can be efficiently updated iteratively during the sampling process. Shampoo (Gupta et al., 2018; Anil et al., 2021) shows how to do this in the context of optimization, deriving a preconditioner possessing a close relationship with full-matrix Adagrad (Duchi et al., 2011). They additionally provide details for a version related to full-matrix RMSprop. We build on that version.

We denote the tensor rank of the $l$th parameter as $d_l$, and the individual shapes of the $l$th parameters as $\{(n_l)_1, \ldots, (n_l)_{d_l}\}$. We can then expressed the metric as

$$G(\boldsymbol{\theta}_t) = \text{diag}(\{\otimes_{i=1}^{d_l}[H_t^i]_l^{\frac{1}{2d_l}}\}_{l=1}^L),$$

$$G(\boldsymbol{\theta}_t)^{-1} = \text{diag}(\{\otimes_{i=1}^{d_l}[H_t^i]_l^{-\frac{1}{2d_l}}\}_{l=1}^L),$$

$$G(\boldsymbol{\theta}_t)^{-\frac{1}{2}} = \text{diag}(\{\otimes_{i=1}^{d_l}[H_t^i]_l^{-\frac{1}{4d_l}}\}_{l=1}^L),$$

where we use $i$ to index the dimension of the parameter, $[H_0^i]_l = \epsilon I_{D_l}$ and

$$[H_t^i]_l = \lambda[H_{t-1}^i]_l + (1-\lambda)[\hat{g}_t]_l^{(i)}.$$

Here $g_{j,j'}^{(i)} := \sum_{\alpha_{-i}} g_{j,\alpha_{-i}} g_{j',\alpha_{-i}}$, where $\alpha$ are all possible indexes; see Appendix A.9 for details on the notation. Each $[H_t^i]_l$ is therefore a square matrix, with both the width and the height given by the $i$th dimension of the shape of the $l$th parameter. We used a slight abuse of notation here, by using $[\hat{g}_t]_l$ to denote the gradients with respect to the corresponding parameter but reshaping them here to match the shape of the parameter that is now a tensor.

Next, we will show how this metric can be used in the context of SGRLD. In the original Shampoo, each $[H_t^i]_l^{-\frac{1}{2d_l}}$ is calculated numerically after adding a small constant $\epsilon$ to avoid numerical issues. For SGRLD we also need access to $[H_t^i]_l^{-\frac{1}{4d_l}}$, and hence we first compute $[H_t^i]_l^{-\frac{1}{4d_l}}$ numerically, and then compute $[H_t^i]_l^{-\frac{1}{2d_l}}$ by taking the square. The Kronecker product formulation allows efficient updates with standard tensor operations (Anil et al., 2021).

The computational cost is then $C(\sum_{l=1}^L \sum_{m=1}^{d_l}(n_l)_m^3)$ and the memory cost is $C(\sum_{l=1}^L \sum_{m=1}^{d_l}(n_l)_m^2)$, for some positive constants $C$. These costs are strictly larger than those of the diagonal metrics or the Monge metric, but in practice the metric can be updated relatively efficiently (e.g. periodically, after hundreds of steps), and hence the computation remains manageable. To further reduce computational complexity, we can divide the tensors into smaller blocks and treat them as individual tensors instead (Anil et al., 2021).

The Shampoo metric also relates to the empirical Fisher information discussed by Kunstner et al. (2019). As shown in Appendix A.8, it has a connection with the square root of the empirical Fisher matrix and it hence avoids the inverse gradient scaling problem, but lacks direct interpretation in terms of curvature. The effect of taking a square root of the metric is further discussed by Staib et al. (2020).

The following theorem shows the asymptotic validity of the sampler, with the proof in the Appendix A.3. The bound depends on the EMA parameter $\lambda$ in the same way as the bound for the Monge metric and otherwise follows that of Theorem 2.1.

**Theorem 3.2.** *For SGRLD with the Shampoo metric, after ignoring the $\Gamma(\boldsymbol{\theta})$ terms during discretized updates, we can bound the approximation error as defined in Theorem 2.1 as*

$$E\left(\hat{\phi} - \bar{\phi}\right)^2 \leq C\left(\sum_t \frac{h_t^2}{S_T^2}E\|\Delta V_t\|^2 + \frac{1}{S_T} + \frac{(\sum_{t=1}^\top h_t^2)^2}{S_T^2}\right) + O(\tau^2(1-\lambda)^2).$$

## 4 Experiments

These experiments characterise the behaviors of the proposed samplers and additionally provide information about the nature of the posteriors of some network architectures.

### 4.1 Experimental setup

**Comparison methods** We evaluate the two proposed metrics against other commonly-used SGRLD algorithms applicable for arbitrary network structures explained in Section 2. We refer to these methods by their metrics, using *Identity* for Welling & Teh (2011), *RMSprop* for the pSGLD method of Li et al. (2016) and *Wenzel* for Wenzel et al. (2020).

**Experimental details** The code that can be used to reproduce all experiments can be found in `https://github.com/ksnxr/SSGRLDNDM`. Concerning neural network experiments, our implementations for all methods are built on top of Fortuin et al. (2021). For *RMSprop*, *Wenzel* and *Shampoo* we use $\lambda = 0.99$ and $\epsilon = 1e-8$, matching the choices of Fortuin et al. (2022), whereas for Monge we use $\lambda = 0.9$ based on good performance in preliminary experiments. For all methods we select constant learning rate based on performance (log-probability) on a separate validation set, and for Monge we additionally select $\alpha^2$ that controls the metric within the same process. For all methods we use a fixed schedule for learning rate to isolate the effect of the metric. For instance, Zhang et al. (2020) showed that cyclical rates can improve sampling, but since the interaction between such schedules and the underlying metric is unknown, we do not employ them here. We use no data augmentation to isolate its effect on likelihood tempering (Kapoor et al., 2022).

We run the samplers for a total of 400 epochs using a batch size of 100. The first 1000 steps are treated as burn-in, and the actual samples used for evaluation are collected after that with a thinning interval of 100 steps. Each setup is repeated for 10 independent runs. Additional details, like the learning rates for each case, are provided in the Appendix A.4 and A.5.

**Evaluation measures** Except for the funnel experiment where we compare against a known ground truth, we evaluate the samplers from the perspective of predictive accuracy and quantification of predictive uncertainty, following the general observations of Wilson & Izmailov (2020). In particular, we evaluate the log-probability $\log p(y|X)$ and classification accuracy computed as the ensemble average over the posterior samples. We compute both for test data not used during inference or validation, averaging the measures over ten independent runs, and also report the running times. Log-probability is a proper scoring rule (Gneiting & Raftery, 2007) that requires correctly representing the predictive uncertainty and is generally used as the primary metric for evaluating quality of neural networks providing predictions with uncertainty estimates (Lakshminarayanan et al., 2017; Tran et al., 2022) and for comparing alternative inference methods (Zhang et al., 2020). However, we note that it does not directly measure how well the samplers explore the true posterior.

We also use a measure that quantifies the curvature of the posterior. Srinivas et al. (2022) proposed a notion, denoted by $\mathcal{C}(\boldsymbol{\theta})$, that measures the model's curvature with respect to the data distribution by quantifying how much the local space deviates from Euclidean. It can be efficiently computed for arbitrary neural networks and they proposed it for training models of lower curvature, but we use their measure for estimating the average curvature of the regions the samplers visit during inference as the expected value of $\mathcal{C}(\boldsymbol{\theta})$ over the posterior samples.

We use this measure for two purposes. It is primarily used as an approximation of the inference complexity. It is not an exact measure as the samplers are not guaranteed to perfectly sample from the true posterior, but still gives indication of how complex the posterior is for different architectures. We also use the average curvature of different samplers as additional indication of their differences. Lower or higher average curvature does not directly indicate a sampler to be more correct since a poor sampler could equally well be unable to reach areas of high curvature or to get stuck there, but differences between methods are still informative.

Finally, we also measured the expected calibration error (ECE) by Naeini et al. (2015) for the predictions. Rahaman & Thiery (2021) explains that ensemble methods, including sampling-based Bayesian neural networks, are not necessarily good in terms of ECE and hence this evaluation is provided as a complementary evidence in Appendix A.7, rather than as a part of the main results.

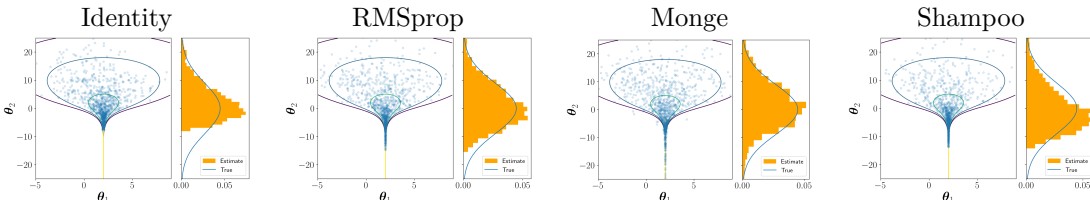

Figure 2: Funnel distribution in four metrics. The better metrics explore the funnel clearly better, though even the best one has difficulties reaching the very end. The right subplots show the challenging marginal of the 2D distribution (left subplot), indicating the true marginal with blue line and the samples with yellow histograms.

## 4.2   Funnel

The funnel distribution $\mathcal{N}(\boldsymbol{\theta}_{1,...,D-1}\,|\,\boldsymbol{\mu}, \mathrm{sp}(\boldsymbol{\theta}_D)I_{D-1})\mathcal{N}(\boldsymbol{\theta}_D\,|\,0, \sigma^2)$, where $\mathrm{sp}(x) = \log(1 + e^x)$, is known to have difficult geometry for sampling and has been used e.g. by Hartmann et al. (2022) and Neal (2003) for sampler evaluation. We consider a funnel distribution with $D = 2$. In Figure 1 we already used this distribution for visual demonstration of how the Monge metric captures the local curvature of the target distribution. In the following, we show how SGRLD algorithms with identity, RMSprop, Monge and Shampoo metrics behave in this task. When the gradient is a vector of one dimension, as it is here, ignoring technicalities like periodic updates and the role of $\epsilon$, Shampoo metric coincides with full matrix RMSprop.

We draw $2e^6$ samples from the funnel, using well chosen hyperparameters for each method. The gradients – that here are known analytically – available for the samplers are corrupted by noise of standard deviation of one, to emulate a scenario where stochastic gradients are used. Figure 2 shows that the identity metric is unable to reach the area of high curvature. The better metrics clearly help, but even the Monge metric that is here the best may not be sampling from the exact correct distribution. Hartmann et al. (2022) showed that LMC can solve the problem nearly perfectly in this metric in combination with exact gradients, but SGRLD is not a perfect algorithm for the problem with stochastic gradients.

## 4.3   Fully-connected networks

We use fully connected neural networks of size 784-$N$-$N$-10 on MNIST dataset (LeCun et al., 2010), where we use $N \in [400, 800, 1200]$, with setup inspired by Li et al. (2016) and Korattikara et al. (2015). We evaluate all the samplers in problems of varying difficulty by (a) changing the prior distribution of the weights and biases and (b) changing the network size by modifying $N$. Tables 1 (log-probability and accuracy) and 2 (average curvature and time) report the results for the commonly used Gaussian prior $\boldsymbol{\theta} \sim \mathcal{N}(\mathbf{0}, \sigma^2 I_D)$ and for the horseshoe prior $\boldsymbol{\theta} \sim \mathcal{N}(\mathbf{0}, \sigma^2 \lambda^2 I_D)$ where $\lambda$ follows half Cauchy distribution with mean 0 and variance 1, as implemented in Fortuin et al. (2021). Following Fortuin et al. (2022), the prior scale $\sigma$ is adjusted based on network structure.

The gist of the results is that one of the non-diagonal metrics is the best in terms of log-probability in all cases, the horseshoe prior is clearly better, and the non-diagonal metrics help more when the posterior is challenging. Even though the numerical differences are somewhat small, the standard deviations (computed over 10 runs) are even smaller and the differences between samplers are reliable. Next we analyse the results in more detail from different perspectives.

**Effect of prior**   For the Gaussian prior the posterior is easy, shown by the low average curvature for all methods. For these cases the choice of the metric is not particularly critical, but the proposed Shampoo metric is still generally the best. For the Monge metric when hidden unit size is 400 or 800 the optimal $\alpha^2 = 0$ and hence the metric reduces to $G(\boldsymbol{\theta}) = I_D$, but with the largest network it reaches an overall best performance with $\alpha^2 = 0.75$.

For the Horseshoe prior the posterior is considerably more challenging, with average curvature around $3 - 4$ times higher, and we see that there is clear benefit in using this prior – the log-probabilities are clearly

| N | Metric | Gaussian prior | | Horseshoe prior | |
|---|---|---|---|---|---|
| | | $\log p(y\|X)$ | **Acc.** | $\log p(y\|X)$ | **Acc.** |
| | Identity | [-0.1310, 0.0004] | [0.9685, 0.0005] | [-0.0750, 0.0004] | [0.9823, 0.0003] |
| | Wenzel | [-0.1315, 0.0002] | [0.9686, 0.0003] | [-0.0768, 0.0009] | [0.9807, 0.0003] |
| 400 | RMSprop | [-0.1307, 0.0005] | [**0.9689**, 0.0004] | [-0.0678, 0.0011] | [0.9798, 0.0008] |
| | Monge | Identity ($\alpha^2 = 0$) is optimal | | [**-0.0629**, 0.0009] | [**0.9831**, 0.0005] |
| | Shampoo | [**-0.1286**, 0.0003] | [**0.9689**, 0.0004] | [-0.0641, 0.0009] | [0.9820, 0.0006] |
| | Identity | [-0.1667, 0.0004] | [0.9587, 0.0003] | [-0.0798, 0.0003] | [0.9818, 0.0004] |
| | Wenzel | [-0.1667, 0.0004] | [0.9583, 0.0005] | [-0.0839, 0.0008] | [0.9789, 0.0003] |
| 800 | RMSprop | [-0.1643, 0.0003] | [**0.9614**, 0.0003] | [-0.0656, 0.0013] | [0.9800, 0.0005] |
| | Monge | Identity ($\alpha^2 = 0$) is optimal | | [-0.0612, 0.0014] | [**0.9834**, 0.0008] |
| | Shampoo | [**-0.1620**, 0.0004] | [0.9605, 0.0003] | [**-0.0603**, 0.0011] | [0.9824, 0.0004] |
| | Identity | [-0.1932, 0.0004] | [0.9514, 0.0004] | [-0.0809, 0.0004] | [0.9814, 0.0002] |
| | Wenzel | [-0.1933, 0.0004] | [0.9519, 0.0003] | [-0.1006, 0.0009] | [0.9746, 0.0005] |
| 1200 | RMSprop | [-0.1886, 0.0005] | [**0.9568**, 0.0003] | [-0.0631, 0.0011] | [0.9808, 0.0007] |
| | Monge | [**-0.1780**, 0.0005] | [0.9560, 0.0005] | [-0.0694, 0.0005] | [0.9833, 0.0004] |
| | Shampoo | [-0.1854, 0.0003] | [0.9559, 0.0004] | [**-0.0564**, 0.0007] | [**0.9834**, 0.0005] |

Table 1: Inference accuracy for MNIST with fully connected networks of varying size $N$ and prior, marking the **best metric** for each configuration. Each entry is given as mean followed by standard deviation.

| N | Metric | Gaussian prior | | Horseshoe prior | |
|---|---|---|---|---|---|
| | | **Curv.** | **Time** | **Curv.** | **Time** |
| | Identity | 5.93 | 8.1 | 14.65 | 8.3 |
| | Wenzel | 5.82 | 7.7 | 14.79 | 9.5 |
| 400 | RMSprop | 6.30 | 8.8 | 25.76 | 10.7 |
| | Monge | $\alpha^2 = 0$ is optimal | | 15.08 | 12.9 |
| | Shampoo | 6.28 | 18.8 | 17.84 | 20.1 |
| | Identity | 4.69 | 19.9 | 15.01 | 24.8 |
| | Wenzel | 4.68 | 20.3 | 14.45 | 26.2 |
| 800 | RMSprop | 4.67 | 24.2 | 20.09 | 29.3 |
| | Monge | $\alpha^2 = 0$ is optimal | | 12.09 | 32.1 |
| | Shampoo | 4.53 | 46.5 | 12.74 | 53.3 |
| | Identity | 4.16 | 43.5 | 15.28 | 56.3 |
| | Wenzel | 4.10 | 49.1 | 11.41 | 58.2 |
| 1200 | RMSprop | 4.11 | 56.9 | 16.96 | 58.3 |
| | Monge | 6.92 | 59.2 | 16.78 | 60.1 |
| | Shampoo | 3.93 | 92.9 | 10.07 | 89.1 |

Table 2: Curvature and computational efficiency for MNIST with fully connected networks of varying size $N$ and prior. The time is given as seconds per epoch.

better than using the Gaussian prior. For this more challenging case, both of the proposed metrics provide substantial improvement over the baselines, especially in terms of log-probability, and the optimal Monge metric always has $\alpha^2 > 0$, with $1.25, 0.5$ and $0.075$ being the optimal values for the three sizes. We also see clear differences between the samplers in terms of the average curvature $\mathcal{C}(\boldsymbol{\theta})$, revealing that the pSGLD sampler using the RMSprop metric behaves in a different manner than the others. For $N = 1200$ with some learning rate the Wenzel metric had high variance over the ten runs, with some bad runs explaining the poor final performance.

**Effect of network size** In terms of the network size the results between the samplers are relatively consistent, but an important observation is that with Gaussian priors the smallest networks are the best whereas with the horseshoe the largest ones outperform the smaller ones – but only if using the better

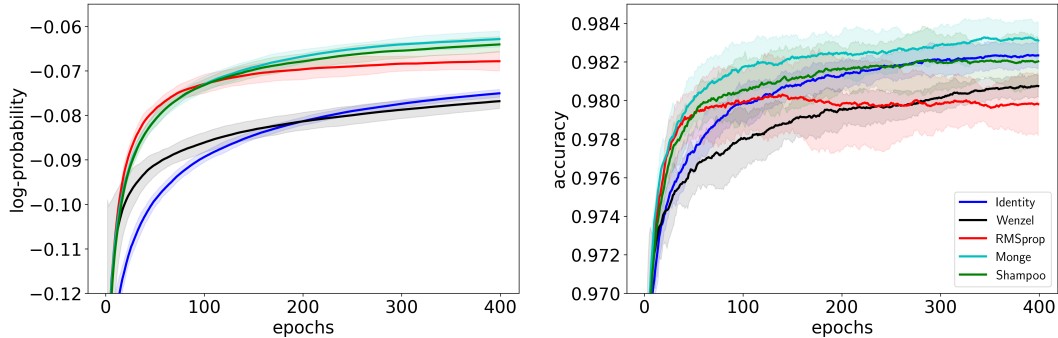

Figure 3: Log-probability (top) and accuracy (bottom) of different samplers on MNIST with hidden layer size 400 and horseshoe prior. Shaded regions show $\pm 1.96$ standard deviations computed over 10 replicates.

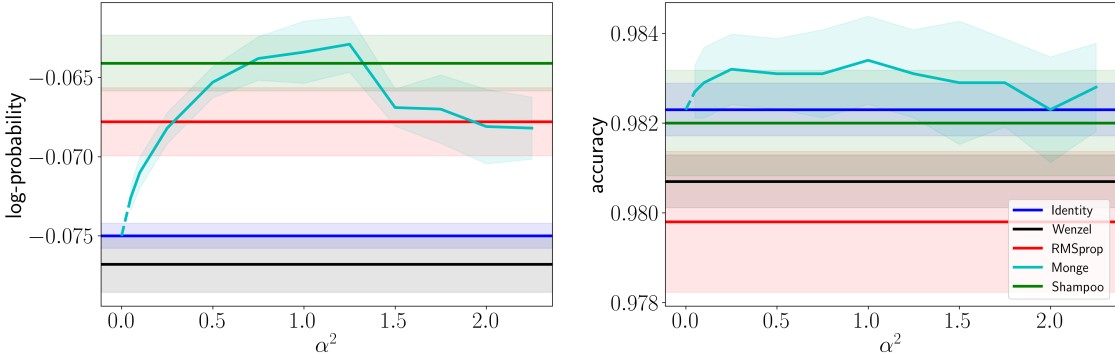

Figure 4: Log-probability (left) and accuracy (right) in Monge metric with varying $\alpha^2$ on MNIST with hidden layer size 400 and horseshoe prior, in comparison to other metrics. Shaded areas show $\pm 1.96$ standard deviations computed over 10 replicates.

samplers. A practitioner using the identity metric might conclude that the best they can do for this data is horseshoe prior with $N = 400$, but in the better metrics we can effectively use also larger networks.

**Convergence** Figure 3 illustrates convergence of the samplers (averaging predictions over samples collected thus far) for the case of $N = 400$ and the horseshoe prior, showing that all metrics result in similar convergence. This plot is shown as a function of the iteration. The Shampoo metric takes around $1.3 - 2.4$ times longer per iteration, but all other metrics share roughly the same cost.

The RMSprop metric that was earlier seen to have clearly different $\mathcal{C}(\boldsymbol{\theta})$ compared to others is here revealed to overfit – it quickly reaches good log-probability but the accuracy does not increase monotonically – confirming that unusual average curvature can be a sign of poor sampling.

**Monge metric parameter** Figure 4 shows how the Monge metric depends on its tuning parameter $\alpha$ for $N = 400$ with the horseshoe prior. For $\alpha^2 = 0$ it matches identity but for $\alpha^2 > 0$ we see clear improvement over the diagonal metric. For $\alpha^2 > 1.25$ the performance starts to degrade possibly due to numerical issues, and the curve suggests it may be possible to further improve the performance if these issues could be better avoided. As noted earlier, in some other cases $\alpha^2 = 0$ is the best choice and the metric becomes identity.

## 4.4 ResNet architecture

As another common network architecture we consider the ResNet (He et al., 2016), known to have relatively easy optimization landscape (Li et al., 2018). We consider two alternative prior choices to inspect the potential effect on the samplers: (a) independent Gaussian priors $\boldsymbol{\theta} \sim \mathcal{N}(0, \sigma^2)$ and (b) correlated normal

| Independent Gaussian prior | | | | |
|---|---|---|---|---|
| **Metric** | $\log p(y\|X)$ | **Acc.** | **Curv.** | **Time** |
| Identity | [-0.4627, 0.0039] | [0.8591, 0.0015] | 10.18 | 7.6 |
| Wenzel | [-0.4869, 0.0047] | [0.8533, 0.0023] | 9.12 | 7.5 |
| RMSprop | [-0.4751, 0.0023] | [0.8566, 0.0014] | 10.27 | 8.8 |
| Shampoo | [**-0.4606**, 0.0043] | [**0.8596**, 0.0018] | 10.63 | 11.4 |
| Correlated normal prior | | | | |
| **Metric** | $\log p(y\|X)$ | **Acc.** | **Curv.** | **Time** |
| Identity | [-0.4437, 0.0040] | [0.8641, 0.0025] | 9.91 | 11.6 |
| Wenzel | [-0.4615, 0.0050] | [0.8614, 0.0021] | 8.91 | 11.8 |
| RMSprop | [-0.4500, 0.0041] | [0.8642, 0.0018] | 9.94 | 14.1 |
| Shampoo | [**-0.4365**, 0.0034] | [**0.8668**, 0.0023] | 10.39 | 15.4 |

Table 3: CIFAR10 with ResNet architecture. The best Monge metric always matches the identity metric and is not shown separately. **Boldface** indicates the best metric. For log-probability and accuracy, each entry is given as mean followed by standard deviation. The time is given as seconds per epoch.

prior proposed by Fortuin et al. (2022). We use the CIFAR10 data (Krizhevsky & Hinton, 2009) and Google ResNet-20 as implemented by Fortuin et al. (2021).

The experimental results are shown in Table 3. Similar to the previous experiments, the Shampoo metric is the best in both cases and the computational overhead over the alternatives is manageable. For the Monge metric the optimal choice is here always to resort to identity metric with $\alpha^2 = 0$ and hence no improvement is observed. In terms of posterior difficulty, we observe that switching to correlated priors makes the curvature slightly lower, matching the hypothesis of Fortuin et al. (2022), while also improving the log-probability and accuracy in terms of all metrics. Shampoo results in the best performances in both cases, while Wenzel and RMSprop are worse than using identity metric. While the standard deviations are larger than the MNIST case, Shampoo still yields consistent improvements over identity especially when using correlated Gaussian prior, with log-probability roughly equal to that of identity metric plus two times the standard deviation.

## 5 Discussion

We used relatively small data and models for evaluation in interest of manageable overall computation. SGLD-based methods have previously been applied on larger problems as well (Zhang et al., 2020), and we showed that the computational overhead of the new metrics is small, with the Monge metric being essentially as fast as the diagonal ones. We see no reason why the methods could not be applied also for larger networks, and we could also take into use various other tricks for maximal scalability, e.g. only doing posterior inference over the last few layers (Lazaro-Gredilla & Figueiras-Vidal, 2010; Ober & Rasmussen, 2019; Watson et al., 2021) or using proper metrics only within individual layers (Gupta et al., 2018; Anil et al., 2021). Our metrics could most likely also be combined with various sampler advances, like the adaptive drift by Kim et al. (2020).

The Shampoo metric performed well in all of our experiments and is currently our practical suggestion. The Monge metric is equally fast as the diagonal ones and in some cases worked extremely well, but requires careful choice of $\alpha^2$ and for the easier posteriors $\alpha^2 = 0$ was actually the best. This is in part caused by numerical stability issues; the method appears in principle to work well with larger values but often has issues with gradient explosion and shows degraded accuracy already before failing completely. This is seen also in Figure 4 where the performance suddenly drops after $\alpha^2 = 1.25$. The likely core reason for these problems is the inverse scaling with the "squared" gradient, which has previously been pointed out as problematic for methods using empirical estimates of Fisher information matrix as preconditioners (Kunstner et al., 2019). It is likely that better normalization or adaptive control of possibly element-wise $\alpha^2$ will resolve these issues, but we do not have good solutions ready yet. The current experiments validate the metric has potential but further development is needed for providing a robust practical method.

We note that in the main experiments we purposefully ignored the samplers' different running times to ensure that they have equal storage cost and computational cost during evaluation, without requiring method-specific thinning intervals or other tricks that would make the interpretations of the results harder. As shown by Tables 2 and 3, the Shampoo metric takes, depending on the case, $1.3 - 2.4$ times longer to compute than the Identity metric. In Appendix A.6 we provide additional empirical results where all samplers are restricted to use the same total computation time, to provide an alternative perspective for sampler efficiency in terms of wall-clock time. We observe that despite some differences, the Monge and Shampoo metrics still retain the advantages over the previous methods.

Our focus was more in demonstrating that it is possible to have computationally efficient metrics that avoid making the diagonal assumption to encourage further research into this direction, instead of arguing that these specific metrics would necessarily be optimal in any specific sense. We used two metrics derived from different perspectives and Lange et al. (2023) recently provided an additional example of constructing a computationally efficient non-diagonal metric specific to the batch normalization operation (Ioffe & Szegedy, 2015), demonstrating that the design space of possible choices is broad. It is likely that other metrics, possibly derived from still different perspectives, can also be made computationally more efficient, but the practical design process remains largely manual.

## 6 Conclusion

Sampling should be done using a metric that somehow accounts for the curvature of the posterior distribution. Current stochastic-gradient samplers usually do this with diagonal metrics to retain computational efficiency and use adaptive metrics motivated by the optimization literature. We showed that it is possible to use non-diagonal metrics while retaining the same or almost the same computational cost, providing two practical metrics that can improve sampling. One metric was consistently good but has some computational overhead. The other worked extremely well in some cases but is sensitive to a tuning parameter, and further research is needed for robust practical implementations.

Despite theoretical advantages, Riemannian samplers are not commonly used today. For instance, none of the leading probabilistic programming environments supports them. In deep learning approximate second-order optimization methods are already widely used and the interest is typically in predictive performance that is easier to evaluate compared to verifying reliability of parameter estimates that can be difficult for Riemannian samplers. This suggests – somewhat counter-intuitively – we might see Riemannian methods entering practice faster in problems of higher dimensionality, now that scalable approaches are available.

Finally, we showed that the need for improved sampling algorithms depends on the model architecture and in particular the prior distribution. Much of the literature has considered networks with simple priors, in particular the Gaussian prior, and for those priors already simple metrics are sufficient. For more advanced priors we can see larger benefits with better metrics.

### Acknowledgments

This work was supported by the Research Council of Finland, by projects 324852, 345811 and 348952 as well as the Flagship programme: Finnish Center for Artificial Intelligence FCAI.

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

## A  Appendix

This Appendix provides proofs for the theorems presented in the paper in Sections A.1-A.3, additional technical details concerning the experiments in Sections A.4-A.5, additional empirical results in Section A.6-A.7, discussions on the relationships of proposed metrics with empirical Fisher in Section A.8 and clarification on the formulation of ShampooSGLD in Section A.9.

### A.1 Convergence analysis

The proof is based on Chen et al. (2015) and Li et al. (2016). We impose the same mild assumption of smoothness and boundedness from their paper.

**Assumption A.1.** $\psi$ and its up to 3rd-order derivatives, $D^k\psi$, are bounded by a function $\mathcal{V}$, i.e. $\|D^k\psi\| \leq C_k$ for $k = (0, 1, 2, 3)$, $C_k, p_k > 0$. Furthermore, the expectation of $\mathcal{V}$ on $\{\boldsymbol{\theta}_t\}$ is bounded: $\sup_t E\,\mathcal{V}^p(\boldsymbol{\theta}_t) \leq \infty$, and $\mathcal{V}$ is smooth such that $\sup_{s\in(0,1)} \mathcal{V}^p(s\boldsymbol{\theta} + (1-s)Y) \leq C(\mathcal{V}^p(\boldsymbol{\theta}) + \mathcal{V}^p(Y))$, $\forall \boldsymbol{\theta}, Y, p \leq \max\{2p_k\}$ for some $C > 0$.

Consider a test function of interest $\phi(\boldsymbol{\theta})$. The posterior average with respect to the invariant measure $\rho(\boldsymbol{\theta})$ related to the Stochastic Differential Equation (SDE) is defined as $\bar{\phi} := \int_{\mathcal{X}} \phi(\boldsymbol{\theta})\rho(\boldsymbol{\theta})d\boldsymbol{\theta}$.

Our geometric MCMC samplers give us samples $(\boldsymbol{\theta}_t)_{t=1}^T$. We approximate $\bar{\phi}$ with $\hat{\phi} := \frac{1}{T}\sum_{t=1}^T \phi(\boldsymbol{\theta}_t)$.

Consider the SDE for SGRLD with exact gradients:

$$d\boldsymbol{\theta} = -G(\boldsymbol{\theta})^{-1}\nabla_{\boldsymbol{\theta}}U(\boldsymbol{\theta})dt + \sqrt{2\tau}G(\boldsymbol{\theta})^{-\frac{1}{2}}dW + \tau\Gamma(\boldsymbol{\theta})dt,$$

its local generator can be written as, where $\mathbf{a} \cdot \mathbf{b} := \mathbf{a}^\top\mathbf{b}$ and $\mathbf{A} : \mathbf{B} := \mathrm{tr}(\mathbf{A}^\top\mathbf{B})$,

$$\mathcal{L}_t = \left(-G(\boldsymbol{\theta}_t)^{-1}\nabla_{\boldsymbol{\theta}_t}U(\boldsymbol{\theta}_t) + \tau\Gamma(\boldsymbol{\theta}_t)\right) \cdot \nabla_{\boldsymbol{\theta}_t} + \tau G(\boldsymbol{\theta}_t)^{-1} : \nabla_{\boldsymbol{\theta}_t}\nabla_{\boldsymbol{\theta}_t}^\top.$$

We define a functional $\psi$ that solves the following Poisson Equation,

$$\mathcal{L}\,\psi(\boldsymbol{\theta}_t) = \phi(\boldsymbol{\theta}_t) - \bar{\phi}. \tag{6}$$

According to the assumptions, $\psi$ exists.

Consider the SDE for Riemannian SGLD after ignoring Gamma:

$$d\boldsymbol{\theta} = -G(\boldsymbol{\theta})^{-1}\nabla_{\boldsymbol{\theta}}\tilde{U}(\boldsymbol{\theta})dt + \sqrt{2\tau}G(\boldsymbol{\theta})^{-\frac{1}{2}}dW,$$

its local generator can be written as

$$\tilde{\mathcal{L}}_t = -G(\boldsymbol{\theta}_t)^{-1}\nabla_{\boldsymbol{\theta}_t}\tilde{U}(\boldsymbol{\theta}_t) \cdot \nabla_{\boldsymbol{\theta}_t} + \tau G(\boldsymbol{\theta}_t)^{-1} : \nabla_{\boldsymbol{\theta}_t}\nabla_{\boldsymbol{\theta}_t}^\top.$$

Let $\mathcal{L}_t$ be the generator of the SDE with exact gradients. Observe that

$$\tilde{\mathcal{L}}_t = \mathcal{L}_t + \Delta V_t + \Delta L_t, \tag{7}$$

where $\Delta V_t = \left(-G(\boldsymbol{\theta}_t)^{-1}(\nabla_{\boldsymbol{\theta}_t}\tilde{U}(\boldsymbol{\theta}_t) - \nabla_{\boldsymbol{\theta}_t}U(\boldsymbol{\theta}_t))\right) \cdot \nabla_{\boldsymbol{\theta}_t}$ and $\Delta L_t = -\tau\Gamma(\boldsymbol{\theta}_t) \cdot \nabla_{\boldsymbol{\theta}_t}$.

We use the Euler-Maruyama integrator, which is a first order integrator. We have

$$E(\psi(\boldsymbol{\theta}_t)) = e^{h_t\tilde{\mathcal{L}}_t}\psi(\boldsymbol{\theta}_{(t-1)}) + O(h_t^2)$$
$$= (\mathbb{I} + h_t\tilde{\mathcal{L}}_t)\psi(\boldsymbol{\theta}_{(t-1)}) + O(h_t^2).$$

Sum over $t$ and plug in Equation 7, we have

$$\sum_{t=1}^T E(\psi(\boldsymbol{\theta}_t)) = \sum_{t=1}^T \psi(\boldsymbol{\theta}_{(t-1)}) + \sum_{t=1}^T h_t\,\mathcal{L}_t\,\psi(\boldsymbol{\theta}_{(t-1)}) + \sum_{t=1}^T h_t\Delta V_t\psi(\boldsymbol{\theta}_{(t-1)}) + \sum_{t=1}^T h_t\Delta L_t\psi(\boldsymbol{\theta}_{(t-1)}) + C\sum_{t=1}^T h_t^2.$$

Divide both side by $S_t$, plug in Equation 6 and re-arrange the terms,

$$\hat{\phi} - \bar{\phi} = \frac{1}{S_T}(E(\psi(\boldsymbol{\theta}_T)) - \psi(\boldsymbol{\theta}_0)) + \frac{1}{S_T}\sum_{t=1}^{T-1}(E(\psi(\boldsymbol{\theta}_t)) - \psi(\boldsymbol{\theta}_t))$$
$$- \frac{1}{S_T}\sum_{t=1}^T h_t\Delta V_t\psi(\boldsymbol{\theta}_{(t-1)}) - \frac{1}{S_T}\sum_{t=1}^T h_t\Delta L_t\psi(\boldsymbol{\theta}_{(t-1)}) + C\frac{\sum_{t=1}^T h_t^2}{S_T}.$$

Therefore, there is a positive constant $C > 0$ satisfying

$$\left(\hat{\phi} - \bar{\phi}\right)^2 \le C\left(\frac{1}{S_T^2}\underbrace{(E(\psi(\boldsymbol{\theta}_T)) - \psi(\boldsymbol{\theta}_0))^2}_{A_1} + \frac{1}{S_T^2}\sum_{t=1}^{T-1}\underbrace{(E(\psi(\boldsymbol{\theta}_t)) - \psi(\boldsymbol{\theta}_t))^2}_{A_2} + \right.$$

$$\left. \sum_{t=1}^{T}\frac{h_t^2}{S_T^2}E\|\Delta V_t\|^2 + \underbrace{\left\|\sum_{t=1}^{T}\frac{h_t}{S_T}\tau\Gamma(\boldsymbol{\theta}_t)\right\|^2}_{A_3} + \left(\frac{\sum_{t=1}^{T}h_t^2}{S_T}\right)^2\right).$$

$A_1$ is bounded according to assumptions, $A_2$ is bounded by $O(\sqrt{h_t})$ because of the added Gaussian noise. Based on these observations, after simplifications, we obtain the following theorem.

**Theorem A.2.** *Denote operator norm as* $\|\cdot\|$. *The MSE of the algorithm can be bounded as*

$$E\left(\hat{\phi} - \bar{\phi}\right)^2 \le C\left(\sum_t \frac{h_t^2}{S_T^2}E\|\Delta V_t\|^2 + \frac{1}{S_T} + \frac{(\sum_{t=1}^{T}h_t^2)^2}{S_T^2} + \left\|\sum_{t=1}^{T}\frac{h_t}{S_T}\tau\Gamma(\boldsymbol{\theta}_t)\right\|^2\right)$$

*for some* $C > 0$.

$A_3$ term can be further relaxed as

$$A_3 \le \left(\sum_{i=1}^{D}\left|\sum_{t=1}^{T}\frac{h_t}{S_T}\tau\Gamma_i(\boldsymbol{\theta}_t)\right|\right)^2.$$

As step sizes are non-zero, it follows that

$$\frac{h_t}{S_T} = O(\frac{1}{T})$$

for any $t$. Since $D$ is bounded by definition, if each $\tau\Gamma_i(\boldsymbol{\theta})$ term is bounded by a relatively small quantity, we can have a reasonably good estimate of $\bar{\phi}$.

## A.2 Derivations for Gamma term in Monge metric

Assume first and second order gradients are bounded. Use the following to simplify notations,

$$\tilde{\nabla}l = \lambda\bar{\nabla}l + (1-\lambda)\hat{\nabla}l,$$

where $\tilde{\nabla}l$ is the moving average after the current update and used to form the metric, $\bar{\nabla}l$ is the moving average before the current update and $\hat{\nabla}l$ is the current stochastic gradients. Further, denote individual entries of the second order derivative for the stochastic estimate of gradients as $\partial_{i,j}^2 l$, or in matrix form as $\nabla^2 l$. Since it is a weighted average of bounded terms, it is also bounded.

The derivations follow that

$$\frac{\partial}{\partial \boldsymbol{\theta}_j}(G(\boldsymbol{\theta})^{-1})_{ij} = \frac{\partial}{\partial \boldsymbol{\theta}_j}(I_D - \alpha^2\frac{\tilde{\nabla}l\tilde{\nabla}l^T}{1+\alpha^2\|\tilde{\nabla}l\|^2})_{ij}$$

$$= -\alpha^2\frac{(1-\lambda)\partial_{i,j}^2 l\tilde{\nabla}l_j + (1-\lambda)\tilde{\nabla}l_i\partial_{j,j}^2 l}{1+\alpha^2\|\tilde{\nabla}l\|^2} + 2\alpha^4\frac{\tilde{\nabla}l_i\tilde{\nabla}l_j(1-\lambda)\sum_{d=1}^{D}\tilde{\nabla}l_d\partial_{d,j}^2 l}{(1+\alpha^2\|\tilde{\nabla}l\|^2)^2},$$

$$\Gamma_i(\boldsymbol{\theta}) = \sum_{j=1}^{D}\frac{\partial}{\partial \boldsymbol{\theta}_j}(G(\boldsymbol{\theta})^{-1})_{ij}$$

$$= -\alpha^2\frac{(1-\lambda)\tilde{\nabla}l^\top(\nabla^2 l)_i + (1-\lambda)\tilde{\nabla}l_i\sum_{j=1}^{D}(\nabla^2 l)_{j,j}}{1+\alpha^2\|\tilde{\nabla}l\|^2} + 2\alpha^4\frac{\tilde{\nabla}l_i\tilde{\nabla}l^\top(1-\lambda)\sum_{j=1}^{D}\tilde{\nabla}l_j(\nabla^2 l)_j}{(1+\alpha^2\|\tilde{\nabla}l\|^2)^2}.$$

Then the $\Gamma(\boldsymbol{\theta})$ term in matrix form can be written as

$$\Gamma(\boldsymbol{\theta}) = (1 - \lambda) \left( -\alpha^2 \frac{\nabla^2 l \tilde{\nabla} l + S_1 \tilde{\nabla} l}{1 + \alpha^2 \|\tilde{\nabla} l\|^2} + 2\alpha^4 \frac{S_2 \tilde{\nabla} l}{(1 + \alpha^2 \|\tilde{\nabla} l\|^2)^2} \right),$$

where

$$S_1 = \sum_{j=1}^{D} (\nabla^2 l)_{j,j},$$

$$S_2 = \tilde{\nabla} l^\top \sum_{j=1}^{D} \tilde{\nabla} l_j (\nabla^2 l)_j.$$

That is, the $\Gamma(\boldsymbol{\theta})$ term for the Monge metric is tractable and can be expressed in matrix form.

Since first order gradients $\nabla l$ are bounded and $\tilde{\nabla} l$ is exponential moving average of $\nabla l$, it follows that $\tilde{\nabla} l$ is also bounded. Since second order gradients are also bounded, we can conclude that $(\nabla^2 l \tilde{\nabla} l + S_1 \tilde{\nabla} l)/(1 + \alpha^2 \|\tilde{\nabla} l\|^2)$ and $S_2 \tilde{\nabla} l/(1 + \alpha^2 \|\tilde{\nabla} l\|^2)^2$ are bounded. Therefore,

$$\Gamma_i(\boldsymbol{\theta}) = O(\alpha^4 (1 - \lambda))$$

for all indexes $i$.

We have for the Monge metric

$$E\left( \hat{\phi} - \bar{\phi} \right)^2 \leq C \left( \sum_t \frac{h_t^2}{S_T^2} E\|\Delta V_t\|^2 + \frac{1}{S_T} + \frac{(\sum_{t=1}^\top h_t^2)^2}{S_T^2} \right) + O(\alpha^8 \tau^2 (1 - \lambda)^2).$$

### A.3 Derivations for Gamma term in the Shampoo metric

To simplify notation, drop dependencies on time and write

$$\tilde{H}^i = \lambda \bar{H}^i + (1 - \lambda) \hat{g}^{(i)},$$

$$G(\boldsymbol{\theta})^{-1} = \mathrm{diag}(\{ \otimes_{i=1}^{d_l} [\tilde{H}^i]_l^{-\frac{1}{2d_l}} \}_{l=1}^L).$$

Observe that every entry in every $\hat{g}^{(i)}$ is a combination of the gradients. Since the gradients are bounded according to assumptions, every entry in every $\hat{g}^{(i)}$ is bounded. Therefore, every $H_{t-1}^i$ is also bounded, and for any indexes $x, y$

$$\frac{\partial}{\partial \boldsymbol{\theta}_k} (H_t^i)_{x,y} = O(1 - \lambda).$$

Recall the Gamma term is,

$$\Gamma_j(\boldsymbol{\theta}) = \sum_{k=1}^{D} \frac{\partial}{\partial \boldsymbol{\theta}_k} (G(\boldsymbol{\theta})^{-1})_{jk}.$$

We have

$$G(\boldsymbol{\theta})_{x,y}^{-1} = \prod_{i=1}^{k} \left( (\tilde{H}^i)^{-\frac{1}{2d}} \right)_{x\%(\prod_{j=1}^i n_j), y\%(\prod_{j=1}^i n_j)},$$

where $x \% n$ is the modulo of $x$ on $n$.

Use $M(\boldsymbol{\theta})$ to denote $G(\boldsymbol{\theta})^{2d}$, then we have

$$M(\boldsymbol{\theta}) = \mathrm{diag}(\{ \otimes_{i=1}^d [\tilde{H}^i]_l \}_{l=1}^L).$$

We can see that each $M(\boldsymbol{\theta})_{jk}$ is the product of $d$ individual terms. It follows that

$$G(\boldsymbol{\theta})^{-1} = M(\boldsymbol{\theta})^{-\frac{1}{2d}},$$

and

$$\frac{\partial}{\partial \boldsymbol{\theta}_k} M(\boldsymbol{\theta})_{jk} = O(1 - \lambda),$$

since it is the sum of product of $O(1 - \lambda)$ and some bounded terms.

Then each $\frac{\partial}{\partial \boldsymbol{\theta}_k}(G(\boldsymbol{\theta})^{-1})_{jk}$ can equivalently be expressed as

$$\frac{\partial}{\partial \boldsymbol{\theta}_k}(M(\boldsymbol{\theta})^{-\frac{1}{2d}})_{jk} = \sum_{l,m,n,o} \frac{\partial(M(\boldsymbol{\theta})^{-\frac{1}{2d}})_{jk}}{\partial(M(\boldsymbol{\theta})^{\frac{1}{2d}})_{lm}} \frac{\partial(M(\boldsymbol{\theta})^{\frac{1}{2d}})_{lm}}{\partial M(\boldsymbol{\theta})_{no}} \frac{\partial M(\boldsymbol{\theta})_{no}}{\partial \boldsymbol{\theta}_k}.$$

$M(\boldsymbol{\theta})$ is a positive definite matrix, therefore $M(\boldsymbol{\theta})^{-\frac{1}{2d}}$ and $M(\boldsymbol{\theta})^{\frac{1}{2d}}$ are also positive definite and unique.

Recall $\partial(X^{-1}) = -X^{-1}(\partial X)X^{-1}$ (Petersen & Pedersen, 2012), so $\frac{\partial(M(\boldsymbol{\theta})^{-\frac{1}{2d}})_{jk}}{\partial(M(\boldsymbol{\theta}))^{\frac{1}{2d}})_{lm}}$ are bounded.

It also holds that (Petersen & Pedersen, 2012)

$$\frac{\partial(X^n)_{kl}}{\partial X_{ij}} = \sum_{r=0}^{n-1}(X^r)_{ki}(X^{n-1-r})_{jl},$$

therefore, $\frac{\partial(M(\boldsymbol{\theta}))^{\frac{1}{2d}})_{lm}}{\partial M(\boldsymbol{\theta})_{no}}$ has a closed-form expression, and is bounded.

To summarize,

$$\frac{\partial(M(\boldsymbol{\theta})^{-\frac{1}{2d}})_{jk}}{\partial(M(\boldsymbol{\theta})^{\frac{1}{2d}})_{lm}} \frac{\partial(M(\boldsymbol{\theta})^{\frac{1}{2d}})_{lm}}{\partial M(\boldsymbol{\theta})_{no}} \frac{\partial M(\boldsymbol{\theta})_{no}}{\partial \boldsymbol{\theta}_k} = O(1 - \lambda),$$

and it trivially follows that

$$\Gamma_j(\boldsymbol{\theta}) = O(1 - \lambda)$$

for all indexes $j$.

Thus, we have for the Shampoo metric

$$E\left(\hat{\phi} - \bar{\phi}\right)^2 \leq C\left(\sum_t \frac{h_t^2}{S_T^2} E\|\Delta V_t\|^2 + \frac{1}{S_T} + \frac{(\sum_{t=1}^{\top} h_t^2)^2}{S_T^2}\right) + O(\tau^2(1 - \lambda)^2).$$

### A.4   Experimental details for sampling from funnel

We use SGRLD samplers with identity, RMSprop, Monge and Shampoon metrics, collecting a total of $2e^6$ samples for each.

Concerning hyperparameters, for identity metric we employ a constant step size of 0.001. For RMSprop metric, we employ a constant step size of 0.0025, $\lambda = 0.995$ and $\epsilon = 0.0$. For Monge metric, we employ constant step size 0.003, $\alpha^2 = 0.1$ and $\lambda = 0.7$. For Shampoo metric, we employ constant step size 0.003, $\lambda = 0.9995$, $\epsilon = 1e - 6$ and update interval of 1.

We observe that when using Monge metric, the sampler might suffer from numerical issues when reaching the bottom of the funnel. When using other metrics, the samplers do not seem to reach those locations. In order to resolve the numerical issue, we resort to identity metric up to a scaling when the norm of the preconditioned gradients is larger than 1000.0.

For the scatter plots, 1000 random samples from all the samples are chosen and plotted.

### A.5   Experimental details for neural network experiments

Following the practice of Wenzel et al. (2020), we use a bijection between learning rate $\ell$ and step size $h$ as $\ell = hn$, and use $\ell$ as the tuning parameter to be validated. We use a separate validation set to tune the

| | | Gaussian prior | Horseshoe prior |
|---|---|---|---|
| $N$ | **Metric** | **Step size** | **Step size** |
| | Identity | 0.05 | 0.25 |
| | Wenzel | 0.075 | 0.75 |
| 400 | RMSprop | 0.00025 | 0.0005 |
| | Monge | Identity ($\alpha^2 = 0$) is optimal | 0.25 |
| | Shampoo | 0.0025 | 0.005 |
| | Identity | 0.05 | 0.5 |
| | Wenzel | 0.05 | 0.5 |
| 800 | RMSprop | 0.0005 | 0.0005 |
| | Monge | Identity ($\alpha^2 = 0$) is optimal | 0.25 |
| | Shampoo | 0.0075 | 0.005 |
| | Identity | 0.025 | 0.25 |
| | Wenzel | 0.05 | 0.25 |
| 1200 | RMSprop | 0.0005 | 0.0005 |
| | Monge | 0.025 | 0.25 |
| | Shampoo | 0.0075 | 0.005 |

Table 4: Learning rates of MNIST experiments

| | Independent Gaussian prior | Correlated normal prior |
|---|---|---|
| **Metric** | **Step size** | **Step size** |
| Identity | 0.1 | 0.1 |
| Wenzel | 0.5 | 0.75 |
| RMSprop | 0.0005 | 0.0005 |
| Shampoo | 0.01 | 0.01 |

Table 5: Learning rates of CIFAR10 experiments

hyperparameters. The learning rates are tuned based on a grid in the form of $[1 \times 10^x, 2.5 \times 10^x, 5.0 \times 10^x, 7.5 \times 10^x]$ for different integers $x$. If in at least one run for a given learning rate a sampler completely breaks down such that it cannot finish sampling, we conclude that the learning rate is not applicable. Concerning $\alpha^2$ for Monge, we first try $\alpha^2 = 1.0$, $\alpha^2 = 0.5$ and $\alpha^2 = 0.1$. If all these choices result in worse or nearly identical performances as identity and the performance generally becomes better as $\alpha^2$ decreases, we conclude that the optimal choice is $\alpha^2 = 0$. Otherwise, we employ grid search in terms of potentially better $\alpha^2$ values.

For fully connected neural networks, following Ghosh et al. (2019), we place horseshoe prior on both the weights and biases.

We report the best learning rates of each sampler in each experimental setup in Table 4 and Table 5.

For the RMSprop metric, following implementations in e.g. Fortuin et al. (2021) and TensorFlow Probability (Dillon et al., 2017), the gradients of the prior are included when calculating the metric. For the Shampoo metric, we employ an update interval of 100 steps.

The running times of the algorithms are estimated by separate 3 runs of the algorithm for 20 epochs each, using a variant of our code base that yields minimal computational workload for training. For MNIST experiments, the code was run on a single Intel® Xeon® Gold 6230 CPU @ 2.10GHz core. For CIFAR10 experiments, the code was run on a single NVIDIA® Tesla® V100-SXM2-32GB GPU with 10 Intel® Xeon® Gold 6230 CPU @ 2.10GHz cores. We measure the time per step, and use that to estimate time per epoch.

### A.6 Experiments for equal computation time

Using non-diagonal metrics typically results in small increase in running time, as observed in Tables 2 and 3. To ensure the slower algorithms are not given an unfair advantage, we here report the results for a setup where

| $N$ | Metric | Gaussian prior | | Horseshoe prior | |
|---|---|---|---|---|---|
| | | $\log p(y|X)$ | **Acc.** | $\log p(y|X)$ | **Acc.** |
| | Identity | [-0.1311, 0.0004] | [0.9684, 0.0005] | [-0.0750, 0.0004] | [0.9823, 0.0003] |
| | Wenzel | [-0.1315, 0.0002] | [0.9686, 0.0003] | [-0.0777, 0.0009] | [0.9806, 0.0003] |
| 400 | RMSprop | [-0.1310, 0.0005] | [**0.9689**, 0.0006] | [-0.0680, 0.0013] | [0.9792, 0.0007] |
| | Monge | Identity ($\alpha^2 = 0$) is optimal | | [**-0.0650**, 0.0006] | [**0.9826**, 0.0006] |
| | Shampoo | [**-0.1304**, 0.0002] | [0.9684, 0.0004] | [-0.0691, 0.0009] | [0.9812, 0.0008] |
| | Identity | [-0.1667, 0.0004] | [0.9587, 0.0003] | [-0.0798, 0.0003] | [0.9818, 0.0004] |
| | Wenzel | [-0.1668, 0.0004] | [0.9583, 0.0004] | [-0.0842, 0.0008] | [0.9787, 0.0002] |
| 800 | RMSprop | [-0.1648, 0.0002] | [**0.9612**, 0.0003] | [-0.0656, 0.0013] | [0.9801, 0.0007] |
| | Monge | Identity ($\alpha^2 = 0$) is optimal | | [**-0.0628**, 0.0010] | [**0.9835**, 0.0007] |
| | Shampoo | [**-0.1635**, 0.0003] | [0.9596, 0.0004] | [-0.0663, 0.0009] | [0.9819, 0.0006] |
| | Identity | [-0.1932, 0.0004] | [0.9514, 0.0004] | [-0.0809, 0.0004] | [0.9814, 0.0002] |
| | Wenzel | [-0.1935, 0.0004] | [0.9519, 0.0003] | [-0.1009, 0.0009] | [0.9745, 0.0005] |
| 1200 | RMSprop | [-0.1894, 0.0006] | [**0.9566**, 0.0003] | [-0.0630, 0.0011] | [0.9808, 0.0007] |
| | Monge | [**-0.1785**, 0.0005] | [0.9560, 0.0005] | [-0.0699, 0.0005] | [**0.9832**, 0.0005] |
| | Shampoo | [-0.1878, 0.0003] | [0.9541, 0.0003] | [**-0.0599**, 0.0005] | [0.9831, 0.0003] |

Table 6: MNIST with fully connected networks of varying size $N$ and prior for setup where each algorithm is constrained to use the same total computation time, marking the **best metric** for each configuration. Each entry is given as mean followed by standard deviation.

each sampler is constrained to have equal total running time, matching the time it took to run the fastest sampler in the main experiments. We note that this may result in worse performance for the samplers which take longer to run than they really are, since they are effectively using fewer samples to form the ensemble, and remark additionally that the exact computation time naturally depends on implementation details and hence not all differences in terms of efficiency are caused by the algorithm details.

In previous experiments, we recorded evaluation metrics after every epoch. Since the time points that the samplers finish running may be different, the best learning rate may also change. We therefore further tune the learning rates of the samplers based on performances on the validation set, assuming that the running times of the samplers under different learning rates are always given by the times reconstructed from the estimated times from before. In most cases the best learning rate coincide with the learning rates as reported in Table 4. The exceptions for fully connected neural networks experiments are: Gaussian prior, $N = 400$, the best learning rate for Shampoo is 0.005; $N = 800$, the best learning rate for Shampoo is 0.0025; $N = 1200$, the best learning rate for Shampoo is 0.0025. Horseshoe prior, $N = 400$, the best learning rate for RMSprop is 0.00075. The exceptions for ResNet experiments are: Gaussian prior, the best learning rate for RMSprop is 0.00075 and the best learning rate for Shampoo is 0.025. Correlated normal prior, the best learning rate for RMSprop is 0.00075 and the best learning rate for Shampoo is 0.025. The final results are again reported for the test set.

Table 6 reports the results for the MNIST data, and in general the results are in line with the main results reported in Table 1. That is, we still have either Monge or Shampoo metric with the best log-probability for all cases, for horseshoe prior also the best accuracy is obtained by the Monge metric in all cases, and the Shampoo metric for the largest network with the horseshoe prior remains among the best overall result.

Table 7 reports the results for the CIFAR10 data. The Shampoo metric that has the slowest per-iteration computation remains as the metric with the best classification accuracy for both priors, but in terms of log-probability the identity metric overtakes all compared methods. However, it is worth noting that it was close to being the best already in the main experiments.

| Independent Gaussian prior | | |
|---|---|---|
| **Metric** | $\log p(y|X)$ | **Acc.** |
| Identity | [**-0.4634**, 0.0039] | [0.8589, 0.0013] |
| Wenzel | [-0.4869, 0.0047] | [0.8533, 0.0023] |
| RMSprop | [-0.4828, 0.0041] | [0.8562, 0.0018] |
| Shampoo | [-0.4858, 0.0047] | [**0.8595**, 0.0023] |
| Correlated normal prior | | |
| **Metric** | $\log p(y|X)$ | **Acc.** |
| Identity | [**-0.4437**, 0.0040] | [0.8641, 0.0025] |
| Wenzel | [-0.4624, 0.0050] | [0.8612, 0.0020] |
| RMSprop | [-0.4594, 0.0030] | [0.8631, 0.0026] |
| Shampoo | [-0.4506, 0.0023] | [**0.8697**, 0.0015] |

Table 7: CIFAR10 with ResNet architecture for setup where each algorithm is constrained to use the same total computation time. The best Monge metric always matches the identity metric and is not shown separately. **Boldface** indicates the best metric. Each entry is given as mean followed by standard deviation.

| $N$ | **Metric** | Gaussian prior
**ECE** | Horseshoe prior
**ECE** |
|---|---|---|---|
| | Identity | [0.0430, 0.0004] | [0.0281, 0.0004] |
| | Wenzel | [0.0438, 0.0004] | [0.0248, 0.0005] |
| 400 | RMSprop | [0.0447, 0.0003] | [0.0124, 0.0009] |
| | Monge | Identity ($\alpha^2 = 0$) is optimal | [0.0180, 0.0009] |
| | Shampoo | [0.0421, 0.0003] | [0.0164, 0.0008] |
| | Identity | [0.0525, 0.0003] | [0.0316, 0.0004] |
| | Wenzel | [0.0513, 0.0003] | [0.0273, 0.0004] |
| 800 | RMSprop | [0.0586, 0.0004] | [0.0106, 0.0004] |
| | Monge | Identity ($\alpha^2 = 0$) is optimal | [0.0178, 0.0009] |
| | Shampoo | [0.0549, 0.0004] | [0.0136, 0.0006] |
| | Identity | [0.0570, 0.0004] | [0.0318, 0.0005] |
| | Wenzel | [0.0581, 0.0002] | [0.0320, 0.0005] |
| 1200 | RMSprop | [0.0689, 0.0004] | [0.0095, 0.0005] |
| | Monge | [0.0546, 0.0004] | [0.0252, 0.0005] |
| | Shampoo | [0.0639, 0.0004] | [0.0114, 0.0006] |

Table 8: ECE for the MNIST experiments

### A.7 Expected Calibration Error

This section complements the empirical results of the main paper by presenting the Expected Calibration Error (ECE) (Naeini et al., 2015) measure for the calibration of the predictions, defined as

$$\text{ECE} = \sum_{i=1}^{K} P(i) \cdot |o_i - e_i|.$$

The predictions are divided into $K$ bins, with $P(i)$ being the empirical probability that predictions fall into the $i$th bin. $o_i$ is the true fraction of positive instances in bin $i$, $e_i$ is the mean of predicted probabilities of being positive in bin $i$. ECE is defined based on the intuition that the level of confidence of a model should be the same as the probability of correct predictions. Principally, lower ECE is better.

The results serve as further verification that the samplers work as intended, while revealing small differences between the samplers. However, we note that there are some criticisms regarding use of ECE in context of deep learning and ensembles, and we refrain from drawing strong conclusions based on these results. ECE is not a proper scoring rule, and can be minimized by a model making uninformative predictions (Ovadia

| Metric | Independent Gaussian prior ECE | Correlated normal prior ECE |
|---|---|---|
| Identity | [0.0919, 0.0032] | [0.0861, 0.0020] |
| Wenzel | [0.1006, 0.0036] | [0.0962, 0.0038] |
| RMSprop | [0.0962, 0.0021] | [0.0925, 0.0021] |
| Shampoo | [0.0902, 0.0017] | [0.0856, 0.0024] |

Table 9: ECE for the CIFAR10 experiments

| Metric | Average confidences |
|---|---|
| Identity | 0.9544 |
| Wenzel | 0.9568 |
| RMSprop | 0.9684 |
| Monge | 0.9648 |
| Shampoo | 0.9663 |

Table 10: Average confidences of MNIST with fully connected neural networks, hidden unit size 400, horseshoe prior

et al., 2019), whereas Rahaman & Thiery (2021) shows that ensembling (as done in sampling-based deep learning methods) is not necessarily improving ECE.

Tables 8 and 9 report the ECE for the MNIST and CIFAR10 experiments, respectively. On MNIST with horseshoe prior that was previously identified as a more challenging sampling task, the RMSprop, Monge and Shampoo samplers result in better ECE compared to Identity and Wenzel. For MNIST with Gaussian prior the differences are generally small. For CIFAR10 experiments Identity and Shampoo have the best ECE.

We additionally ran the samplers for 3 independent runs on MNIST, hidden unit size 400, horseshoe prior, using a variant of our code recording the confidences (the largest predicted probabilities of the model) and report the averages of the confidences in Table 10. These numbers that are approximately 96% for all samplers can be compared with the accuracies that are approximately 98% as indicated in Table 1, confirming the general observation of Rahaman & Thiery (2021); all the samplers are somewhat under-confident.

### A.8 Discussion on the relationship of the metrics with Empirical Fisher

The Fisher information matrix characterises the local curvature of a probabilistic model, and is formed using expectation of the gradients over the possible outcomes of the model. Kunstner et al. (2019) discusses methods that replace the expectation by conditioning on the observed data $y$ instead, resulting in the so-called empirical Fisher

$$\sum_n \nabla_{\boldsymbol{\theta}} \log p(y_n|x_n, \boldsymbol{\theta}) \nabla_{\boldsymbol{\theta}} \log p(y_n|x_n, \boldsymbol{\theta})^\top \tag{8}$$

computed as the sum of gradients for individual data points. They argue it may not be a good choice as a curvature matrix of preconditioner, but contains useful information about the gradient noise in stochastic optimization. Here we describe how both Monge and Shampoo metrics relate to the empirical Fisher, but also have certain important differences. Both metrics use gradients of the joint density rather than the likelihood, but even if ignoring the prior the expressions are different.

The Monge metric (ignoring stochastic estimation of gradients for now) is

$$I_D + \alpha^2 \left( \sum_n \nabla_{\boldsymbol{\theta}} \log p(\boldsymbol{\theta} \,|y_n, x_n) \right) \left( \sum_n \nabla_{\boldsymbol{\theta}} \log p(\boldsymbol{\theta} \,|y_n, x_n) \right)^\top, \tag{9}$$

where the latter term resembles the empirical Fisher but is different: We are now taking the outer product of the full-data gradients, rather than summing over the outer products. Within MongeSGLD we compute

a stochastic estimate of this quantity, but again the expression only considers the observed data $y$ instead of taking the expectation over their distribution. The core difference between the two expressions is that the second term in Equation 9 includes cross-terms between individual samples while Equation 8 does not.

The Shampoo metric has a connection with full matrix RMSprop and the full matrix RMSprop can be interpreted as using an estimate, up to scaling, of

$$\sqrt{\sum_n \nabla_{\boldsymbol{\theta}} \log p(\boldsymbol{\theta} \,|\, y_n, x_n) \nabla_{\boldsymbol{\theta}} \log p(\boldsymbol{\theta} \,|\, y_n, x_n)^{\top}}.$$

That is, Shampoo is related to the empirical estimate for the Fisher matrix as in Equation 8 but takes the square root of that. As discussed by Kunstner et al. (2019), this removes possible scaling issues but makes the connection to Fisher information more vague.

### A.9 Clarification on the formulation of ShampooSGLD

For the convenience of notations, we are using superscript $i$ when introducing the formulation of ShampooSGLD. In expressions like $[H_t^i]_l$, $i$ is not the tensor component, but just an extra index referring to each dimension of the gradient, indexing the matrices. For each $i$, $[H_t^i]_l$ is a square matrix, with both height and width being the shape of the gradients in that dimension. These square matrices are constructed based on the moving average of $[\hat{g}_t]_l^{(i)}$, which is obtained from the gradients. We are using a slight abuse of notation here, because when describing the previous methods we used $\hat{g}_t$ to denote the stochastic gradient concatenated and reshaped to a $D \times 1$ column vector, where $D$ is the total number of parameters; but here we are using $[\hat{g}_t]_l$ to denote the gradients of the $l$th parameter in the same shape of the original parameter.

