# OpenReview forum: "Scalable Stochastic Gradient Riemannian Langevin Dynamics in Non-Diagonal Metrics"
_TMLR — Accepted by TMLR_

### Review · Reviewer_fKj6 · 2023-06-08

**Summary Of Contributions:**

This paper applies two (existing) computationally cheap Riemannian metric methods to Riemannian langevin monte carlo sampling. There is some theory (showing that the metric matters beyond the 0th order term), and some experiments show that the proposed methods better sample from the distribution.

**Audience:**

Yes

**Claims And Evidence:**

Yes

**Requested Changes:**

N/A

**Strengths And Weaknesses:**

Strengths
-----------
* The paper rigorously supports its claims.
* Some experiments are given, but there are some shortcomings (see weaknesses).

Weaknesses
---------------
* Most of the methodology is effectively preexisting. In particular, the metrics effectively come from preexisting literature. Importantly, these metrics were previously tested on much larger scale experiments (e.g. Shampoo was used to train some large text to video diffusion models, if I remember correctly).
* Connected to the above: the experiments are rather small scale and toy. They also really don't show that much improvement (e.g. see horseshoe accuracy numbers). Also, the additional time (e.g. from the shampoo metric) is 2-3x, which is also a rather large burden.
* Why is the Sherman-Morrison Formula intractable here? It has been used to train DEQs (https://arxiv.org/pdf/1909.01377.pdf) for instance, which seem to have much higher dimensionality.

---

> ### Author Response · Authors · 2023-06-19
> **Response to comments**
>
> Thank you for the comments. We uploaded a new version of the paper that addresses the comments of all reviewers, and provide answers to your remarks below.
>
> > Most of the methodology is effectively preexisting. In particular, the metrics effectively come from preexisting literature. Importantly, these metrics were previously tested on much larger scale experiments (e.g. Shampoo was used to train some large text to video diffusion models, if I remember correctly).
>
> Both metrics indeed have been presented before, but non-trivial development was needed to use them efficiently in SGLD as detailed below:
>
> - The Monge metric has previously been used only in very small-scale problems, with no previous attempts of combining it with stochastic gradients or neural networks in any way. Prior to our work it was not at all obvious the algorithm can be implemented for deep networks in linear time, and Theorem 3.1 was needed to validate the algorithm. The empirical observation that further work is needed to make it robust across different models and tasks is also valuable for the research community, even though naturally a bit of a disappointment for us.
>
> - Shampoo was previously used only in optimization. Even though the extension to sampling is mathematically relatively straightforward, both the theoretical validity (Theorem 3.2) and empirical evaluation were missing. The improvement was smaller than we hoped for, but nevertheless the metric is consistently the best and often by a margin of more than two standard deviations.
>
> > Connected to the above: the experiments are rather small scale and toy. They also really don't show that much improvement (e.g. see horseshoe accuracy numbers). Also, the additional time (e.g. from the shampoo metric) is 2-3x, which is also a rather large burden.
>
> We intentionally chose small-scale experimentation so that the experiments are easier to reproduce and understand for readers not familiar with more complex models, and saw now reason run large models just for the sake of it. As you mentioned, Shampoo has already been used for larger models even though it has basically the same computational overhead also in the optimization case, which to an extent counters the argument of the extra cost being a large burden.
>
> > Why is the Sherman-Morrison Formula intractable here?
>
> The sentence was written in a bit confusing way. We do use Sherman-Morrison and the purpose of the sentence was just to remind the reader that for very large models it is still costly to form the whole $D\times D$ metric and perform the necessary computation. Instead, we need to manipulate the expressions to a format where we do not need to explicitly construct the matrix. We now revised the paragraph to explain this better.

---

### Review · Reviewer_jecC · 2023-06-09

**Summary Of Contributions:**

The paper proposes two different approaches for using non-diagonal metrics with stochastic gradient samplers. The first approach utilizes a Monge metric, which approximates the Fisher information matrix as the identity plus a low-rank matrix. The second approach exploits Kronecker factors, as in Shampoo, to approximate the Fisher information. Both approaches efficiently support key operations like inversion and square roots. The authors empirically demonstrate that these proposed approaches can yield improvements on MNIST-MLP and CIFAR10-RESNET benchmarks in terms of log-likelihood and accuracy with reasonable computational overhead.

**Audience:**

Yes

**Broader Impact Concerns:**

I don't have any concerns on the ethical implications of the work.

**Claims And Evidence:**

Yes

**Requested Changes:**

- In the introduction, authors claim that there is no stochastic-gradient sampler with non-diagonal $G(\boldsymbol{\theta})$ for training large neural network. It would be helpful to clarify why methods such as [1] do not fall into this category in the manuscript. At least for the models the authors experiment on, I believe that [1] can be employed.
- For Funnel problem in Section 4.2, I anticipate that Shampoo-based methods would be more beneficial in capturing the correlations. However, I could not find the Shampoo results and it would be helpful to include it.
- The accuracies reported in Table 1 for all methods (including the baseline) seems to be lower than I would anticipate [2, 3]. Could the authors clarify why this might be the case? This also applies to the results presented in Table 2 .

[1] Nado, Z., Snoek, J., Grosse, R., Duvenaud, D., Xu, B., & Martens, J. (2018). Stochastic gradient Langevin dynamics that exploit neural network structure.

[2] Blundell, C., Cornebise, J., Kavukcuoglu, K., & Wierstra, D. (2015, June). Weight uncertainty in neural network. In International conference on machine learning (pp. 1613-1622). PMLR.

[3] Fortuin, V., Garriga-Alonso, A., Ober, S. W., Wenzel, F., Rätsch, G., Turner, R. E., ... & Aitchison, L. (2021). Bayesian neural network priors revisited. arXiv preprint arXiv:2102.06571.


**Strengths And Weaknesses:**

Strengths
- The paper is well-written with clear motivation and an extensive overview of related methods.
- The experimental setup is justified. I especially appreciated the toy example in Section 4.2 that shows the strengths of incorporating non-diagonal information.
- The authors included code and experiment details in the Appendix.

Weakness
- The improvements for incorporating non-diagonal information seem to be limited. For example, Monge always prefers $\alpha = 0$ for CIFAR10 experiments and it seems like extensive tuning for $\alpha$ is necessary.
- There are some minor concerns on the experiments section, which I address below.

---

> ### Author Response · Authors · 2023-06-19
> **Response to comments**
>
> Thank you for the comments. We uploaded a new version of the paper that addresses the comments of all reviewers, and provide answers to your remarks below.
>
> > The improvements for incorporating non-diagonal information seem to be limited. For example, Monge always prefers for CIFAR10 experiments and it seems like extensive tuning for is necessary.
>
> The Monge metric indeed does not help always, but in the case of MNIST with horseshoe prior is the best method while having no computational overhead. We were naturally hoping for bigger and more consistent improvement, but this empirical finding is valuable as such as indication that the metric has potential but further work is needed for making it generally applicable.
>
> > In the introduction, authors claim that there is no stochastic-gradient sampler with non-diagonal $G(\boldsymbol{\theta})$ for training large neural network. It would be helpful to clarify why methods such as [1] do not fall into this category in the manuscript. At least for the models the authors experiment on, I believe that [1] can be employed.
>
> The method of [1] (Nado et al. 2018) indeed uses a non-diagonal metric but in our opinion cannot be considered a general approach for *arbitrary* deep neural networks, since it is somewhat limited in scope and difficult to apply in practice. The original publication provides the mathematical details only for fully connected networks and Anil et al. (2021) (Appendix B) points out difficulties in implementing many commonly used operations like batch norm and layer norm. We now clarified the sentence to avoid misleading the readers.
>
>
> > For Funnel problem in Section 4.2, I anticipate that Shampoo-based methods would be more beneficial in capturing the correlations. However, I could not find the Shampoo results and it would be helpful to include it.
>
> When the gradient is a one-dimensional vector, as in the funnel case, the metric estimated by Shampoo is (ignoring technical details like periodic updates and the role of $\epsilon$) the same as full matrix RMSprop, as briefly discussed in Section 4.2. Hence, the plot is also similar to that of RMSprop. We originally left it out to save space, but now included the funnel plot also for Shampoo for completeness. It works better than identity, but not as well as Monge.
>
> > The accuracies reported in Table 1 for all methods (including the baseline) seems to be lower than I would anticipate [2, 3]. Could the authors clarify why this might be the case? This also applies to the results presented in Table 2.
>
> The accuracies are lower than in [2, 3] because we did not use some of the heuristics considered in those works. For example, [3] (Fortuin et al., 2022) used cyclical learning rates that improve the accuracy, and we explained in Section 4.1 why we did not use them in our experiments. We sincerely believe that the specific effect of a metric is best communicated for the readers in context of an algorithm that is otherwise as standard as possible.

---

### Review · Reviewer_3sW2 · 2023-06-10

**Summary Of Contributions:**

This submission proposes a method for sampling from approximate
Bayesian posterior distributions using on stochastic Langevin dynamics.
The main contribution is two new approximate curvature matrices
which can be estimated online and used to improve
sample quality and the exploration ability of the sampler.
The first approach, called the Monge metric, uses a diagonal plus rank-one matrix,
where the rank-one component is estimated online using stochastic gradients
of the log-probability.
The second, called Shampoo metric, is based on a Kronecker factored curvature
matrix from the optimization literature.
The authors characterize the sample efficiency of these two approaches
and evaluate their practical performance of MNIST and CIFAR-10.

**Audience:**

Yes

**Broader Impact Concerns:**

None.

**Claims And Evidence:**

Yes

**Requested Changes:**

I would like to see calibration experiments evaluating the quality of uncertainty for the the neural network experiments. I think such experiments are critical to judging the quality of the posterior approximations provided by the sampling methods. Using non-diagonal curvature matrices should lead to improved sampling performance and improved posterior uncertainties. Otherwise, non-diagonal metrics are not worth the additional computational complexity. See Yao et al. [2] for a basic reference on evaluating calibration for approximate inference methods.



**Strengths And Weaknesses:**

### Strengths and Weaknesses

This is an interesting submission which attempts to improve stochastic-gradient-based
posterior samplers using non-diagonal metrics.
Non-diagonal metrics can better capture the curvature of the log-probability,
leading to improved exploration and higher sample efficiency in downstream
tasks.
As such, I think this paper is of interest to the machine learning
community and a good fit for TMLR.

The major strength of this submission is the relative simplicity of the
non-diagonal metrics, which are based on well-known approximate curvature
matrices from the optimization literature, and
This means fast implementations are already available and can be leveraged
immediately to develop better samplers.

However, the submission has several significant weaknesses. From a theoretical
perspective, the approximate curvature matrices seem to worsen the sample complexity
of estimation compared to using the identity matrix (Theorems 3.1 and 3.2).
Moreover, the novelty/relevance of these theorems are clear to me from
the submission, although I am not an expert in this area.

Empirically, the Shampoo metric does lead to improved log-likelihood, but
essentially the same accuracy as using the identity matrix on both MNIST
and CIFAR-10.
Thus, it's not clear these improvements are meaningful in practice.
Moreover, no calibration experiments are provided, so its impossible to
judge the quality of the uncertainty provided by the samples.
Log-likelihoods are not a suitable metric for judging calibration, since they
encourage posterior collapse (the MAP estimate maximizes predictive log-likelihood).
Thus, I find the experimental evaluation to be weak at best.

Finally, the utility of the Monge metric is not well justified by the experiments
or theory.
On CIFAR-10 and MNIST with a Gaussian prior, the curvature matrix
defaulted to identity ($\alpha^2 = 0$), so it's not clear why Monge is useful
in general.
This matches Theorem 3.1, where the complexity suggests $\alpha^2 = 0$
is optimal.
Finally, is well-known that the outer-product of gradient is a poor estimate
of the Fisher information [1], so the justification here seems very weak.
Perhaps this is why $\alpha^2 = 0$ is often optimal.

To summarize:

**Strengths**:

- The non-diagonal metrics are fast to compute, improve the log-likelihood on most tasks,
    and well-known in optimization.
- The paper is well-written and polished.


**Weaknesses**:

- The theory seems relatively weak and doesn't justify using non-diagonal metrics.
- No calibration experiments are provided for MNIST/CIFAR-10, so it's unclear
    non-diagonal metrics lead to improved uncertainty estimation.
- The Monge metric often reverts to the identity matrix, introduces an additional
    tuning parameter, and isn't well-justified.

### Correctness

As mentioned, this paper is outside of my area of research and makes use
of tools I am not familiar with, so I cannot comment on correctness of the
theoretical results.
I did not check the derivations in the appendix for correctness.

### Writing and Presentation:

The paper is well-written and I noticed essentially no typographical issues.
I congratulate the authors on producing such a well-polished text.


### Additional References:


[1] Kunstner, Frederik, Philipp Hennig, and Lukas Balles. "Limitations of the empirical Fisher approximation for natural gradient descent." Advances in neural information processing systems 32 (2019).

[2] Yao, Yuling, et al. "Yes, but did it work?: Evaluating variational inference." International Conference on Machine Learning. PMLR, 2018.

### Questions

- Theorem 3.1: So the claim here is that increasing $\alpha$ reduces $\mathbb{E} \|\Delta V_t\|^2$?
    Otherwise, it seems like the tightest bound in 3.1 comes from setting $\alpha = 0$ and using the diagonal metric.
    I think a precision definition of $\Delta V_t$ should be included in in the statement so we can see this trade-off, if it exists.
    Currently the text only says "where $\Delta V_t$ is an operator" (Theorem 2.1), which isn't helpful.

- Section 3.2: I don't understand your notation. I see that $l$ indexes layers of the model,
    and $i$ indexes the up to the rank of the tensor; $[H_{t}^i]_l$ is ith
    tensor component, but what is its shape (is it actually a tensor or just a matrix)
    and how do you update it with the gradient?
    Its not clear to me how your "slight abuse of notation" works.
    Also, how do you take the Kronecker product of a $d_i$ tensors?
    That should produce a tensor whose shape is the product of the component shapes, right?

- Theorem 3.1 and 3.3: What does "ignoring the $\Gamma(\theta)$ terms mean?
    Does it mean dropping the $\Gamma$ term in Eq. (3), which you say can be done safely
    for practical algorithms?
    Or does it mean dropping the last term in Theorem 2.1 from the analysis?

- Can you provide calibration information for fully connected networks? Log-likelihood is
    not a good substitute for calibration information, since the MAP estimate,
    which incorporates no uncertainty information, will maximize the log-likelihood.
    Thus, log-likelihoods typically favor posterior collapse around the MAP and
    don't capture uncertainty information very well.

- Table 3: Monge reduces to identity, which seems to match Theorem 3.1.
    It doesn't seem like choosing $\alpha^2 > 0$ is supported by any strong
    evidence---empirical or theoretical.
    Can you comment on this?

---

> ### Author Response · Authors · 2023-06-19
> **Response to comments / part 1**
>
> Thank you for the detailed feedback. We uploaded a new version of the paper that addresses the comments of all reviewers and provide responses for your remarks below, starting with the requested changes.
>
> > I would like to see calibration experiments evaluating the quality of uncertainty for the the neural network experiments. I think such experiments are critical to judging the quality of the posterior approximations provided by the sampling methods. Using non-diagonal curvature matrices should lead to improved sampling performance and improved posterior uncertainties. Otherwise, non-diagonal metrics are not worth the additional computational complexity. See Yao et al. [2] for a basic reference on evaluating calibration for approximate inference methods.}
>
> As will be detailed below, the log-probability is a standard metric for evaluating sampling methods, quantifying also the predictive uncertainty which is the more relevant quantity for (neural network) models with non-identifiable parameterization. We now extended the justification of the evaluation measure and its properties in Section 4.1. In addition, we now provide in Appendix A.7 the Expected Calibration Error (ECE) for the samplers as another measure of quality of the predictive uncertainty.
>
> Yao et al. [2] proposes good methods for evaluating the quality of distributional approximations (variational and Laplace approximations), but they are not applicable for evaluation of MCMC methods as they are either based on importance sampling that requires the approximation in closed form, or require a point estimate which is hard to define due to non-identifiability. In brief, their methods unfortunately cannot be used here.
>
> > From a theoretical perspective, the approximate curvature matrices seem to worsen the sample complexity of estimation compared to using the identity matrix (Theorems 3.1 and 3.2). Moreover, the novelty/relevance of these theorems are clear to me from the submission
>
> The theorems provide an upper bound for the approximation error for a sampler that ignores the computationally difficult $\Gamma(\boldsymbol{\theta})$ term. The general Theorem 2.1 was provided by Li et al. (2016), but Theorems 3.1 and 3.2 are contributions of this work and critical for ensuring the validity of the samplers -- if the error was not bounded we could not justify ignoring the term. The proofs follow the same pattern as that of Li et al. and are not major theoretical contributions, but they are non-trivial and necessary for our work.
>
> Note that there is no reason to believe the bounds to be tight, and hence we cannot conclude that non-zero $\alpha$ would necessarily increase the approximation error.
>
> > Can you provide calibration information for fully connected networks? Log-likelihood is not a good substitute for calibration information, since the MAP estimate, which incorporates no uncertainty information, will maximize the log-likelihood. Thus, log-likelihoods typically favor posterior collapse around the MAP and don't capture uncertainty information very well.
>
> First, we clarify that our main interest is in quantifying the predictive uncertainty of the model, not the posterior approximation itself. This is a common perspective in Bayesian deep learning (Wilson and Izmailov, 2020), where the posterior is non-identifiable. Log-likelihood is a proper scoring rule and hence it captures also the notion of uncertainty for the predictive distribution. It is often considered the primary metric for quantifying the uncertainty in Bayesian deep learning (see e.g. Lakshminarayanan et al. (2017) and Tran et al. (2022)).
>
> We remind that besides accuracy and log-likelihood we also characterized the average curvature of the obtained samples during the sampling chain (Section 4.1 and Tables 2 and 3). To further extend the empirical validation we now report also the expected calibration error (ECE) measure of Naeini et al. (2015) in the Appendix, validating that the proposed metrics result in similar or improved calibration of predictive uncertainty across all scenarios. However, we still consider the log-likelihood as the primary metric.
>
> > Finally, the utility of the Monge metric is not well justified by the experiments or theory. On CIFAR-10 and MNIST with a Gaussian prior, the curvature matrix defaulted to identity ($\alpha^2 = 0$), so it's not clear why Monge is useful in general. This matches Theorem 3.1, where the complexity suggests is optimal.
>
> The Monge metric is indeed not yet robust and general, but for MNIST with horseshoe prior it is overall the best metric while having no computational overhead. This demonstrates the potential value, and by presenting the sampler we encourage follow-up work that improves the generality e.g. by suitable adaptation of $\alpha^2$. As noted before, we also believe there is value in explicitly showing that further work is needed to make it generally applicable. We now re-worded the conclusions to make this clear.

---

> ### Author Response · Authors · 2023-06-19
> **Response to comments / part 2**
>
> >Finally, is well-known that the outer-product of gradient is a poor estimate of the Fisher information [1], so the justification here seems very weak. Perhaps this is why $\alpha^2 = 0$ is often optimal.
>
> This is an excellent remark and an opportunity to make an important clarification. We now discuss the relationship between the Monge and empirical Fisher in Section 3.1, with some additional details in Appendix A.8. The empirical Fisher as discussed by [1] (Kunstner et al. 2019) is formed by taking the sum of the outer product of gradients, where each gradient considers a single data point only, whereas the term in the Monge metric the outer product of the full gradient, where the gradient now is the sum of gradients comprising a subset of the whole data set.
>
> In effect, the empirical Fisher is an expectation of outer products whereas the term in Monge metric is outer product of expectations. Despite somewhat similar looking equations they behave very differently, and it is important to clarify the difference. Empirical Fisher is indeed biased, may not lead to stable numerical inversion, and does not work as a preconditioner as nicely shown by [1].
> The second term in the Monge metric, however, is an unbiased stochastic estimate for the Fisher when the target distribution is a likelihood, it captures cross-terms not included in the empirical Fisher, and it was demonstrated to work well as preconditioner in earlier work in other context.
>
> That said, we agree that the proposed metric shares the scaling issue of Empirical Fisher, due to inverse scaling with the squared gradient magnitude. This is possibly a major reason for the lack of improvement over $\alpha^2=0$ in some cases, and we now mention this in Sections 3.1 and 5.
>
> > Theorem 3.1: So the claim here is that increasing $\alpha$ reduces $E\vert\Delta V_{t}\vert^{2}$? Otherwise, it seems like the tightest bound in 3.1 comes from setting $\alpha = 0$ and using the diagonal metric.  I think a precision definition of $\Delta V_{t}$ should be included in in the statement so we can see this trade-off, if it exists.
>
> The definition of $\Delta V_{t}$ is provided in Appendix A.1, right below Equation (7), and intuitively it is an operator caused by using stochastic gradients rather than exact gradients. We did not include it in the main paper because it requires substantial amount of additional notation that is only relevant for the Appendix and introducing all of that in the main paper would make the paper unnecessarily difficult to read, but we now mention explicitly that it is provided in the Appendix.
>
> As mentioned above and in Section 2, the theorems prove that the samplers are valid in the asymptotic limit. They cannot be used for justifying the choice of $\alpha^2$ since we naturally only collect a finite sample and the bounds are likely not tight.
>
> > Section 3.2: I don't understand your notation. I see that $l$ indexes layers of the model, and $i$ indexes the up to the rank of the tensor; $[H_{t}^{i}]_{l}$ is ith tensor component, but what is its shape (is it actually a tensor or just a matrix) and how do you update it with the gradient? Its not clear to me how your "slight abuse of notation" works. Also, how do you take the Kronecker product of a $d_i$ tensors? That should produce a tensor whose shape is the product of the component shapes, right?
>
> OpenReview support for Latex formatting is a bit limited and made clarifying the notation difficult in the response, but we added a clarifying paragraph as Appendix A.9 in the new version. The abuse of notation is because $\hat{g}_{t}$ previously meant concatenated stochastic gradient reshaped to a column vector of $D$ dimensions, whereas here the subscript $l$ refers to a part of that in the shape of the original parameter.
>
> > Theorem 3.1 and 3.2: What does "ignoring the $\Gamma(\boldsymbol{\theta})$ terms mean? Does it mean dropping the $\Gamma(\boldsymbol{\theta})$ term in Eq. (3), which you say can be done safely for practical algorithms? Or does it mean dropping the last term in Theorem 2.1 from the analysis?
>
> It refers to dropping it in Eq. (3). The term is computationally costly, but the theorems prove that we get a valid sampler even when ignoring it. The analysis includes the term.

---

> > ### Comment · Reviewer_3sW2 · 2023-06-25
> >
> > Thanks for responding to my review.
> >
> > > As mentioned above and in Section 2, the theorems prove that the samplers are valid in the asymptotic limit. They cannot be used for justifying the choice of  since we naturally only collect a finite sample and the bounds are likely not tight.
> >
> > I think a statement similar to this should be included in the main paper. If the authors don't think that the bounds are tight enough to give useful information about the hyper-parameters, then the text should emphasize that the theorems are only useful for showing asymptotic validity. After Theorem 3.1, the text says "We can clearly see the role of $\alpha$ in the above bound," but I disagree with this and find it a bit misleading given that the authors' are now arguing that the real role of $\alpha$ is not captured by the $\alpha^8$ term.
> >
> >  > It refers to dropping it in Eq. (3). The term is computationally costly, but the theorems prove that we get a valid sampler even when ignoring it. The analysis includes the term.
> >
> > Please clarify this in the paper.
> >
> > > ... is an unbiased stochastic estimate for the Fisher when the target distribution is a likelihood
> >
> > It isn't obvious to me that the outer-product of expectations is an estimator of the expectation of the outer-product under the model, which is the typical definition of the Fisher information. Maybe the authors can add their comments on this to the paper and provide a reference for unbiasedness, which they neglected to do here.
> >
> >
> > > We now report also the expected calibration error (ECE) measure of Naeini et al. (2015) in the Appendix
> >
> > You should include the definition of ECE in the appendix as well. Right now these experiments don't provide any information to readers (including me) who don't know how ECE is defined or why it is a good measure of calibration.
> >
> >
> > > First, we clarify that our main interest is in quantifying the predictive uncertainty of the model, not the posterior approximation itself. This is a common perspective in Bayesian deep learning (Wilson and Izmailov, 2020), where the posterior is non-identifiable. Log-likelihood is a proper scoring rule and hence it captures also the notion of uncertainty for the predictive distribution. It is often considered the primary metric for quantifying the uncertainty in Bayesian deep learning (see e.g. Lakshminarayanan et al. (2017) and Tran et al. (2022)).
> >
> > Do you agree that the MAP estimate, which has no notion of predictive uncertainty since it is a point estimator, maximizes the predictive log-likelihood and so dominates all other models in this metric? If yes, how can you argue that log-likelihood is the right metric for measuring predictive uncertainty? I'm aware that many authors use log-likelihood to evaluate the quality of the posterior predictive distribution for BNNs, but I disagree that this is the right metric for the task.

---

> > > ### Author Response · Authors · 2023-06-27
> > > **Response to comments**
> > >
> > > Thanks for your further comments. We incorporated the clarifications regarding the roles of the theorems and the description of ECE to the manuscript and just uploaded a new version. Responses and descriptions of changes relating to the other comments are provided below.
> > >
> > > > It isn't obvious to me that the outer-product of expectations is an estimator of the expectation of the outer-product under the model, which is the typical definition of the Fisher information. Maybe the authors can add their comments on this to the paper and provide a reference for unbiasedness, which they neglected to do here.
> > >
> > > After reconsideration, we decided to take out the claims relating to bias of the estimates and both Section 3.1 and Appendix A.8 now only explain how the term in the Monge metric differs from the empirical Fisher. The detailed analysis of the two estimates required a bit too elaborate notation and would benefit also from empirical experiments to quantify their properties, and it is well beyond the scope of this paper where the main contribution is ensuring the previously proposed metric can be implemented efficiently for this class of models.
> > >
> > > The previous answer was also incorrect, in the sense that we cannot currently ensure the Monge term is unbiased estimate of Fisher when conditioning on the observed $y$. We had verified that the expectation of that term over the distribution of the outputs equals the Fisher information under the assumption of independent observations, but this does not yet say anything about the bias. We do not elaborate this further in the paper now that we removed all claims regarding the bias.
> > >
> > > > Do you agree that the MAP estimate, which has no notion of predictive uncertainty since it is a point estimator, maximizes the predictive log-likelihood and so dominates all other models in this metric?
> > >
> > > We would like to apologize for answering a bit besides the point, since use of proper scoring rules relates to capturing the distribution of the predictions, and indeed not about uncertainty of the parameters.  You are correct that MAP estimate can have high log-likelihood and hence log-likelihood in general case does not say anything about whether the uncertainty over the parameters was correctly captured. We are not aware of any general results stating MAP estimates would dominate over other estimators (for instance, maximum likelihood dominates MAP for poorly chosen priors), but that specific point is not particularly relevant for our paper.
> > >
> > > We slightly revised Section 4.1 to ensure that there is no risk of confusion regarding test log-likelihood quantifying the posterior quality directly, but instead the chosen measure is motivated by the previous work.

---

> > > > ### Comment · Reviewer_3sW2 · 2023-06-27
> > > >
> > > > Thanks for replying and updating the manuscript. I'm largely in agreement with your comments and I think my concerns have been addressed.
> > > >
> > > > > We are not aware of any general results stating MAP estimates would dominate over other estimators (for instance, maximum likelihood dominates MAP for poorly chosen priors), but that specific point is not particularly relevant for our paper.
> > > >
> > > > Agreed. My point is only that the MAP estimate will have the maximum training log-likelihood out of all inference methods using the same prior distribution. This holds by definition of the MAP estimate! If the problem is "easy", then the MAP estimate can have the best test log-likelihood as well, which shows how the test log-likelihood can fail as a measure of the quality of the posterior predictive distribution. That is my main point.

---

### Decision · Action_Editors · 2023-07-13

**Recommendation:** Accept as is

**Comment:**

This paper presents two methods for performing Stochastic Gradient Riemannian Langevin Dynamics in larger scale deep networks, where the curvature is modeled using something more sophisticated than a simple diagonal matrix. The methods include a diagonal plus rank-one update and a KFAC-like Shampoo based sampler.  Drawing stochastic samples from large deep networks is certainly of interest to many within the community, for uncertainty quantification, generative modeling, Bayesian deep learning, etc.  The reviewers felt that the claims were all supported by the evidence and all of them agreed on acceptance.  There were some questions regarding treatment of previous work (e.g. I agree that KFAC-SGLD and the Shampoo-based method are rather similar).  I would ask the authors to tone down the wording regarding this comparison in their paper (as they promised in the discussion period).  Given the consensus of the reviewers, the recommendation is accept the paper.

**Audience:**

Drawing samples from large scale deep networks is an active area of research and interesting to communities exploring uncertainty quantification, generative modeling and Bayesian deep learning.

**Claims And Evidence:**

The reviewers are all in agreement regarding claims and evidence - i.e. that the claims are sufficiently supported.